# Mitochondrial complexome reveals quality-control pathways of protein import

Uwe Schulte[1,2,12], Fabian den Brave[3,12], Alexander Haupt[1,12], Arushi Gupta[3,4,12], Jiyao Song[3,4], Catrin S. Müller[1], Jeannine Engelke[3], Swadha Mishra[3], Christoph Mårtensson[4,9], Lars Ellenrieder[4,10], Chantal Priesnitz[4], Sebastian P. Straub[4,5,11], Kim Nguyen Doan[4], Bogusz Kulawiak[4,6], Wolfgang Bildl[1], Heike Rampelt[2,4], Nils Wiedemann[2,4,7], Nikolaus Pfanner[2,4,7 ✉], Bernd Fakler[1,2,8 ✉] & Thomas Becker[3]

Mitochondria have crucial roles in cellular energetics, metabolism, signalling and quality control[1–4]. They contain around 1,000 different proteins that often assemble into complexes and supercomplexes such as respiratory complexes and preprotein translocases[1,3–7]. The composition of the mitochondrial proteome has been characterized[1,3,5,6]; however, the organization of mitochondrial proteins into stable and dynamic assemblies is poorly understood for major parts of the proteome[1,4,7]. Here we report quantitative mapping of mitochondrial protein assemblies using high-resolution complexome profiling of more than 90% of the yeast mitochondrial proteome, termed MitCOM. An analysis of the MitCOM dataset resolves >5,200 protein peaks with an average of six peaks per protein and demonstrates a notable complexity of mitochondrial protein assemblies with distinct appearance for respiration, metabolism, biogenesis, dynamics, regulation and redox processes. We detect interactors of the mitochondrial receptor for cytosolic ribosomes, of prohibitin scaffolds and of respiratory complexes. The identification of quality-control factors operating at the mitochondrial protein entry gate reveals pathways for preprotein ubiquitylation, deubiquitylation and degradation. Interactions between the peptidyl-tRNA hydrolase Pth2 and the entry gate led to the elucidation of a constitutive pathway for the removal of preproteins. The MitCOM dataset—which is accessible through an interactive profile viewer—is a comprehensive resource for the identification, organization and interaction of mitochondrial machineries and pathways.

Mitochondria are multifunctional organelles. In addition to their roles in oxidative phosphorylation and in the metabolic pathways of amino acids, lipids, haem and iron–sulfur clusters, they perform functions in cellular signalling, redox processes, quality control and apoptosis[1–3]. Most mitochondrial proteins are imported as precursors from the cytosol, whereas about 1% of the proteins are synthesized inside the organelle. Mitochondria are dynamic organelles that frequently divide and fuse, and they display a characteristic folded structure of their inner membrane. Defects in the mitochondria can lead to severe diseases, particularly of the central nervous system, metabolism and the cardiovascular system[2,3].

The protein complement of the mitochondria has been determined in systematic proteomic studies[3–5], with a coverage of more than 90% for the mitochondrial proteome in the model organism baker's yeast (*Saccharomyces cerevisiae*)[6]. By contrast, the organization of mitochondrial proteins into protein assemblies, from stable complexes and supercomplexes to transient assembly intermediates, is only partially understood. Various approaches such as affinity purification, native electrophoresis, gel filtration, density gradients, cross-linking and structural biology have been applied to study the organization of the mitochondrial proteome[1,4,7]. Each of these approaches provided important information on selected mitochondrial complexes, but none yielded a comprehensive overview of the expected large number of distinct protein assemblies.

We report a comprehensive high-resolution complexome of yeast mitochondria and associated proteins, based on blue native electrophoresis combined with cryo-slicing and mass spectrometry analysis (csBN–MS)[8,9]. We systematically improved csBN–MS from protein separation to advanced detection and quantification of protein profiles, yielding the high resolution and coverage of the mitochondrial complexome (MitCOM)[7,9,10]. MitCOM covers more than 90% of the high-confidence mitochondrial proteome and provides a wealth of information on mitochondrial protein assemblies and their quantitative appearance.

[1]Institute of Physiology, Faculty of Medicine, University of Freiburg, Freiburg, Germany. [2]CIBSS Centre for Integrative Biological Signalling Studies, University of Freiburg, Freiburg, Germany. [3]Institute of Biochemistry and Molecular Biology, Faculty of Medicine, University of Bonn, Bonn, Germany. [4]Institute of Biochemistry and Molecular Biology, ZBMZ, Faculty of Medicine, University of Freiburg, Freiburg, Germany. [5]Faculty of Biology, University of Freiburg, Freiburg, Germany. [6]Laboratory of Intracellular Ion Channels, Nencki Institute of Experimental Biology, Polish Academy of Sciences, Warsaw, Poland. [7]BIOSS Centre for Biological Signalling Studies, University of Freiburg, Freiburg, Germany. [8]Center for Basics in NeuroModulation, Freiburg, Germany. [9]Present address: MTIP, Basel, Switzerland. [10]Present address: Novartis, Basel, Switzerland. [11]Present address: Sanofi-Aventis (Suisse), Vernier, Switzerland. [12]These authors contributed equally: Uwe Schulte, Fabian den Brave, Alexander Haupt, Arushi Gupta. ✉e-mail: nikolaus.pfanner@biochemie.uni-freiburg.de; bernd.fakler@physiologie.uni-freiburg.de

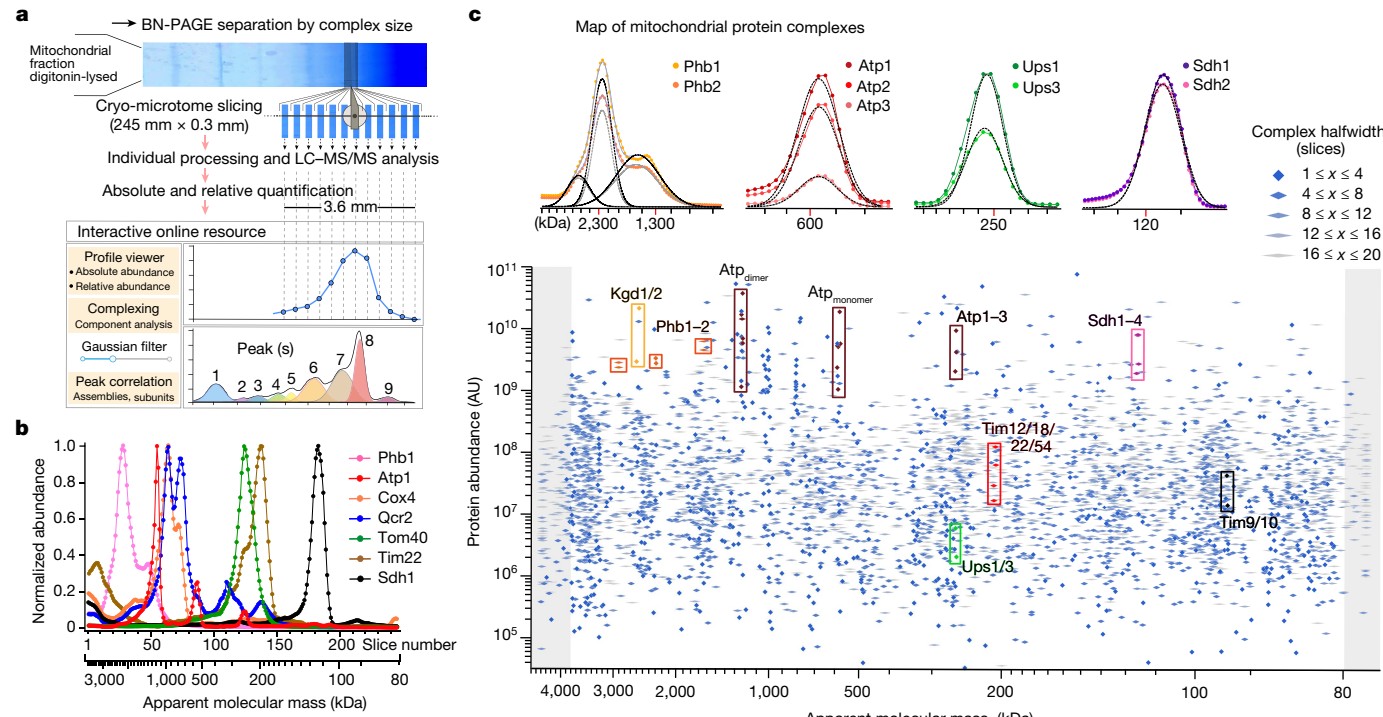

**Fig. 1 | High-resolution complexome profiling of mitochondria using improved csBN–MS. a**, Schematic of the csBN–MS workflow, comprising protein separation by blue native gel electrophoresis (BN-PAGE), cryo-slicing of the gel, MS analysis, data processing and protein quantification. The interactive online platform presents all resulting abundance–mass profiles and MS data for user-directed inspection and evaluation. LC–MS/MS, liquid chromatography coupled with tandem MS. **b**, Abundance profiles over slice number and apparent molecular mass for the indicated set of mitochondrial proteins; the profiles were normalized to the maximal abundance value determined for the respective protein. **c**, Map of protein peaks determined from abundance–mass profiles by multi-Gaussian fits and verified by manual inspection. All of the symbols represent identified peak parameters (apparent molecular mass, abundance and half-width as indicated at the top right). The shaded areas denote peaks with maxima outside the blue native gel. Coloured frames denote known protein complexes that are assembled from the indicated constituents of which the peaks are highlighted in the same colour. Insets: sections of abundance–mass profiles determined for the indicated protein constituents of established mitochondrial assemblies together with the fit lines (dashed) obtained by Gaussian fitting. For Phb1 and Phb2, the composite fit lines (solid, blue) and the individual components (dashed, grey, black) are shown. Note the close match in the midpoint of the individual Gaussian functions obtained in independent fits for the constituents of the same complex.

## High-resolution mitochondrial complexome

Yeast mitochondria were lysed with the non-ionic detergent digitonin and processed for blue native gel electrophoresis, high-resolution cryo-slicing and quantitative MS (Fig. 1a and Extended Data Figs. 1 and 2). We optimized the conditions for protein separation, quantification and evaluation to substantially improve the resolution and proteome coverage, including (1) preparation and lysis of mitochondria under mild conditions, followed by separation of protein assemblies by blue native electrophoresis; (2) cryo-slicing of blue native gel lanes into 245 slices with a width (0.3 mm) three times smaller than the half-width of the sharpest-focusing protein peaks; and (3) elaborate MS quantification and profile building (Extended Data Fig. 1 and Methods). We determined abundance–mass profiles for 1,891 different proteins, 906 of them represented high-confidence mitochondrial proteins and made up around 96% of the protein mass in our preparation[6] (MitCOM; Supplementary Tables 1 and 2), whereas another 985 profiles originated from non-mitochondrial proteins representing about 4% of the total protein mass (Extended Data Fig. 2a and Supplementary Table 3). Mit-COM covers a molecular mass range from around 80 kDa to 3,800 kDa with homogenous resolution and a protein abundance range of more than six orders of magnitude (Fig. 1b,c, Extended Data Fig. 2c,d and Supplementary Tables 1 and 2). The individual abundance–mass profile of each MitCOM protein represents a protein-specific fingerprint with a characteristic shape for the distribution and abundance of this protein in one or more assemblies (peaks) (Fig. 1b and Extended Data Fig. 2b,e).

We performed an unbiased automated component analysis based on peak detection and the fitting of multi-Gaussian functions to the abundance–mass profiles, followed by rigorous quality control by manual inspection. We identified a combined 5,224 peaks in the profiles of the 818 MitCOM proteins accessible to the fitting procedure (Fig. 1c and Extended Data Fig. 2e). The number of identified protein peaks substantially exceeds previous complexome profiling efforts in mitochondria[7,9,11]. The MitCOM dataset is presented in an openly accessible platform for interactive analysis through the integrated profile viewer (https://www.complexomics.org/datasets/mitcom or https://www3.cmbi.umcn.nl/cedar/browse/experiments/CRX36; Extended Data Fig. 3). The resolution and accuracy of the MitCOM profiles promoted unsupervised systemic analysis of assemblies and their composition, a major challenge in the exploration of native protein complexes[11,12]. A distance measure combining information on peak intensity, mass range and correlation coefficients (Extended Data Fig. 4 and Methods) was used for the automated discrimination of protein complex components by $t$-distributed stochastic neighbourhood distribution ($t$-SNE) (Extended Data Figs. 4 and 5).

## Complexity of protein organization

An analysis of the MitCOM dataset showed that the majority of mitochondrial proteins appeared as constituents of several assemblies with an average of 6.4 peaks per protein (complexity), whereas only about 1% of the proteins exhibited a single peak (Fig. 2 and Extended Data

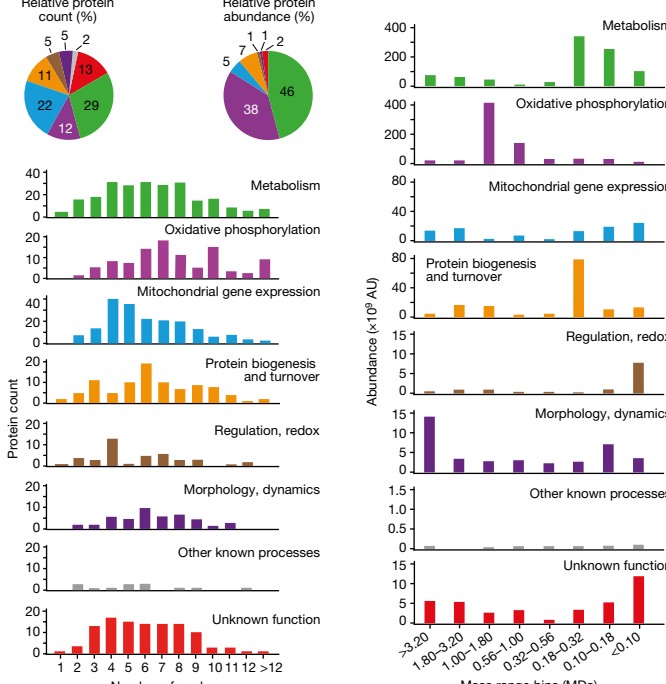

**Fig. 2 | Complexity and size of mitochondrial protein complexes related to functional categories.** Summary of the complexity (left, number of peaks per protein) or the apparent molecular size (right, mass range bins) of assemblies/complexes identified in the abundance–mass profiles of 815 proteins assigned to distinct categories of mitochondrial function[6] (Extended Data Fig. 6). The relative numbers (left) and abundances (right) of the functional categories are shown. The apparent mass range was binned into equal log-value intervals (bins of 0.25) of the apparent molecular mass.

Fig. 6a). The complexity was largely independent of protein properties such as the number of predicted transmembrane segments and the molecular mass, and only moderately correlated with protein abundance (Extended Data Fig. 6b). However, complexity as well as molecular mass ranges of protein assemblies markedly differed between the four mitochondrial subcompartments—the outer membrane, intermembrane space, inner membrane and matrix (Extended Data Fig. 6c). The functional classification of MitCOM protein assemblies revealed characteristic differences (Fig. 2). The highest mean number of peaks per protein was observed for the subcategories cofactor biosynthesis (for example, coenzyme Q biosynthesis[4,6]) and oxidative phosphorylation assembly (Extended Data Fig. 6d). The molecular mass distribution of protein assemblies differed substantially between the functional categories such as metabolism, oxidative phosphorylation, protein biogenesis, regulatory processes and morphology (Fig. 2). Proteins with an as yet unknown function, representing only around 2% of the MitCOM mass but around 13% of the different MitCOM proteins, were found in low- and high-molecular-mass ranges. Thus, small as well as very large mitochondrial protein assemblies contain a considerable number of proteins that lack functional annotation.

The non-mitochondrial proteins in the source preparation showed an average complexity of 4.5 peaks per protein and predominantly originated from cellular compartments in the vicinity of mitochondria, cytosol, endoplasmic reticulum and nucleus (Extended Data Figs. 2a and 7 and Supplementary Table 3). Protein biogenesis, degradation and quality control represented the largest functional category, consistent with the identification of quality-control factors associated with mitochondria outlined below.

Taken together, the MitCOM dataset uncovered a notable complexity of mitochondrial protein assemblies over an extended molecular mass and abundance range, suggesting the occurrence of many as yet unknown protein complexes and interactions of this organelle.

## Dynamic protein assemblies

MitCOM demonstrated high precision in the analysis of dynamic protein machineries such as the outer membrane sorting and assembly machinery (SAM), and the separation of complexes of different abundance, such as the carrier translocase of the inner membrane (TIM22) and succinate dehydrogenase (SDH, respiratory complex II) (Extended Data Fig. 8a–c). Similarly, MitCOM quantitatively separated distinct complexes and subcomplexes of the $F_1F_0$-ATP synthase (Extended Data Fig. 8d).

We examined whether MitCOM can be used to identify unknown binding partners of protein assemblies. A population of the outer membrane protein Om14 was found to co-migrate with the translocase of the outer membrane (TOM) complex at around 250 kDa, whereas another population of Om14 migrated together with its established interaction partner Om45 in a low-molecular-mass assembly[13,14] (Fig. 3a). We verified the interaction between Om14 and TOM using two-step affinity purification, which yielded a highly purified TOM complex containing Om14 but not Om45 (Fig. 3b). Om14 was reported to function as a receptor for translating cytosolic ribosomes[15]; however, it has been unclear how Om14-bound ribosomes can transfer nascent precursor polypeptides to the mitochondrial entry gate TOM. MitCOM shows that Om14 is present in two populations, Om14–Om45 and Om14–TOM, revealing the missing direct link of Om14 to the protein import site. Thus, the ribosome receptor can cooperate with TOM in co-translational protein import.

To study whether MitCOM can reveal new interaction partners of very large assemblies, we analysed inner membrane complexes. We identified the yeast J protein Mdj2 as a partner of the prohibitin–*m*-AAA supercomplex of about 2.2 MDa (Extended Data Fig. 8e,f). By comparing the MitCOM profiles of assembly factors of cytochrome *c* oxidase (COX, respiratory complex IV), we found that a main peak of Shy1 is present at the fully assembled $III_2IV_2$ respiratory supercomplex of 1 MDa, different from other assembly factors such as Mss51 and Coa1 (Extended Data Fig. 8g,h). We conclude that the high resolution and quantitative nature of MitCOM make it a powerful source for the analysis of dynamic protein machineries and the identification of interaction partners.

## Pth2 is involved in the removal of preproteins

We investigated whether MitCOM can be used to identify pathways that involve dynamic interactions and components of low abundance, such as quality-control pathways that monitor mitochondrial proteostasis and remove accumulated preproteins[16–20] (Extended Data Fig. 9a). We observed a co-migration of the peptidyl-tRNA hydrolase Pth2 with the TOM complex (Fig. 3c). Pth2 is a mitochondrial outer-membrane protein[6,21]. In addition to its peptidyl-tRNA hydrolase activity[22,23], Pth2 interacts with cytosolic factors of the ubiquitin proteasome system[24]. It has been unknown whether mature Pth2 has mitochondrial interaction partners and whether it has a role in mitochondrial function or quality control.

Affinity purification through tagged Tom40 as well as tagged Pth2 demonstrated the association between Pth2 and the TOM complex (Extended Data Fig. 9b,c). Deletion of the *PTH2* gene caused a growth defect on non-fermentable medium on which mitochondrial respiration is critical for cell survival, and a double-mutant strain *pth2Δpam17Δ* displayed a synthetic growth defect (Extended Data Fig. 9d). Pam17 is a subunit of the presequence translocase-associated motor (PAM) of the mitochondrial protein import machinery[25,26]. Loss of Pam17 reduces the protein import efficiency and causes an accumulation of mitochondrial preproteins when the degradation of non-imported proteins is inhibited[17]. The precursor form of Mdj1 accumulated in *pth2Δpam17Δ* cells (Fig. 3d), indicating that the removal of non-imported preproteins

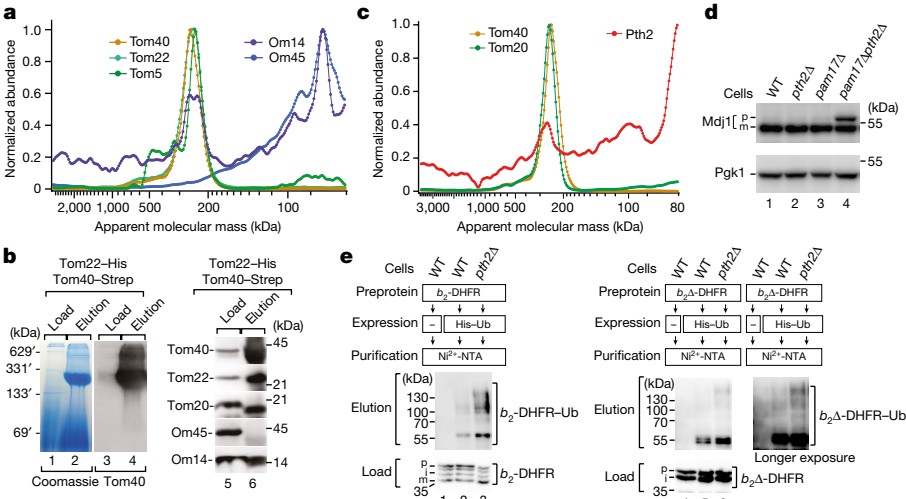

**Fig. 3 | The TOM complex interacts with the ribosome receptor Om14 and the quality-control factor Pth2. a**, Normalized abundance–mass profiles of the TOM subunits Om14 and Om45. **b**, Tom22–His Tom40–Strep mitochondria were lysed with digitonin and processed for tandem affinity purification through Ni-NTA agarose and Strep-Tactin Sepharose. The samples were analysed by blue native electrophoresis or SDS–PAGE followed by Coomassie staining (lanes 1 and 2) or immunodetection using the indicated antisera. Load, 0.2% (blue native gel) or 1% (SDS–PAGE); elution, 100%. **c**, Normalized abundance–mass profiles of Tom40, Tom20 and Pth2. **d**, Cell extracts of the indicated strains were analysed by immunodetection. p, precursor; m, mature form. **e**, Wild-type (WT) and $pth2\Delta$ cells expressing cytochrome $b_2$–DHFR or $b_2\Delta$–DHFR precursors and His-tagged ubiquitin as indicated were lysed under denaturing conditions and affinity-purified through Ni-NTA agarose. Proteins were analysed by SDS–PAGE and immunodetection. Affinity purification of tagged ubiquitin under denaturing conditions leads to an enrichment of proteins with covalently attached ubiquitin[41]. $b_2$-DHFR–Ub, ubiquitin modified $b_2$-DHFR; $b_2\Delta$-DHFR–Ub, ubiquitin modified $b_2\Delta$-DHFR; i, intermediate.

was impaired. To analyse whether accumulated preproteins were ubiquitylated, we used a yeast strain expressing tagged ubiquitin and cytochrome $b_2$ precursor variants containing dihydrofolate reductase (DHFR) (Methods). In $pth2\Delta$-mutant cells, increased amounts of ubiquitylated preproteins accumulated (Fig. 3e). Thus, the lack of Pth2 leads to an accumulation and ubiquitylation of mitochondrial preproteins, suggesting that Pth2 is involved in the removal of preproteins from the import-site TOM.

## Precursor ubiquitylation at the TOM complex

The accumulation of ubiquitylated preproteins at the mitochondrial entry gate[17,27] (Fig. 3e) and the ubiquitylation-mediated modulation of mitochondrial protein biogenesis in mammalian cells[27,28] suggest that ubiquitylation and deubiquitylation are important in the removal of preproteins from TOM. However, neither an E3 ubiquitin ligase nor a deubiquitylase operating at the yeast TOM complex have been identified. We extended the target-focused screening of MitCOM to proteins that are potentially involved in ubiquitylation and deubiquitylation. Out of several candidate proteins, populations of the cytosolic HECT-type E3 ubiquitin ligase Rsp5 ('reverses $spt^-$ phenotype') and of the outer-membrane-localized ubiquitin-specific protease 16 (Ubp16) co-migrated with the TOM complex (Fig. 4a). Affinity purification with tagged TOM subunits confirmed the interactions of Rsp5 and Ubp16 with TOM (Fig. 4b). Several functions have been assigned to Rsp5, including regulation of biosynthesis of unsaturated fatty acids, ubiquitylation of endoplasmic reticulum–mitochondria contact sites, and the fusion GTPase Fzo1, protein degradation in the cytosol and multivesicular body formation[29–32]. Ubp16 is a predicted, yet lacking characterization, yeast deubiquitylase (homologue of USP30) with an amino-terminal anchor in the outer mitochondrial membrane[21,33].

We analysed whether Rsp5 and Ubp16 are involved in the removal of mitochondrial preproteins. Deletion of *PAM17* in the *rsp5-1* yeast mutant resulted in a synthetic growth defect and accumulation of the Mdj1 precursor (Fig. 4c and Extended Data Fig. 9e), indicating that Rsp5 has a role in the removal of non-imported preproteins. Using yeast

with tagged ubiquitin, we observed ubiquitylated $b_2$-DHFR and Mdj1 precursors. Ubiquitylation of both precursors was blocked in *rsp5-1* cells (Fig. 4d). As control, loss of Mdm30 and Mfb1, which are F-box protein subunits of SCF ubiquitin ligase complexes that regulate mitochondrial morphology[34,35], did not impair the ubiquitylation of the precursors (Fig. 4d). Thus, Rsp5 is required for the ubiquitylation of mitochondrial preproteins at the TOM complex (Extended Data Fig. 9f–h).

In mutant cells lacking the predicted deubiquitylase Ubp16, ubiquitylated forms of the Mdj1 and $b_2\Delta$-DHFR precursors accumulated (Fig. 4e–g and Extended Data Fig. 10a,b). By combining $ubp16\Delta$ with the *rsp5-1* mutant, the ubiquitylation of Mdj1 and $b_2\Delta$-DHFR was blocked (Fig. 4f,g), demonstrating that ubiquitylation by Rsp5 was a prerequisite for subsequent deubiquitylation of accumulated preproteins by Ubp16. The E3 ubiquitin ligases Doa10 and Hrd1 were, like Mdm30 and Mfb1, dispensable for the ubiquitylation of accumulated $b_2\Delta$-DHFR (Extended Data Fig. 10c).

We conclude that Rsp5 and Ubp16 function in the ubiquitylation and deubiquitylation of mitochondrial preproteins at the TOM complex. In the $pth2\Delta ubp16\Delta$ double mutant, the ubiquitylation of $b_2\Delta$-DHFR was enhanced further (Fig. 4g). Together with the accumulation of ubiquitylated preproteins in $pth2\Delta$-mutant cells (Fig. 3e), this indicates that Pth2 is involved in the removal of ubiquitylated precursors.

## Pth2 and mitochondrial quality control

We examined whether Pth2 is part of an already known quality-control pathway of mitochondria or whether it constitutes a separate pathway for the removal of preproteins. First, the mitochondrial protein translocation-associated degradation (mitoTAD) pathway removes non-imported preproteins accumulated at the TOM complex under constitutive conditions, whereas the mitochondrial compromised protein import response operates during stress conditions[16,17]. Pth2 is expressed and interacts with the TOM complex under constitutive conditions (Fig. 3c and Extended Data Fig. 9b) like Ubx2, a core component of mitoTAD[17]. Ubx2 binds to TOM and recruits the AAA-ATPase Cdc48, thereby promoting the release of translocation-arrested

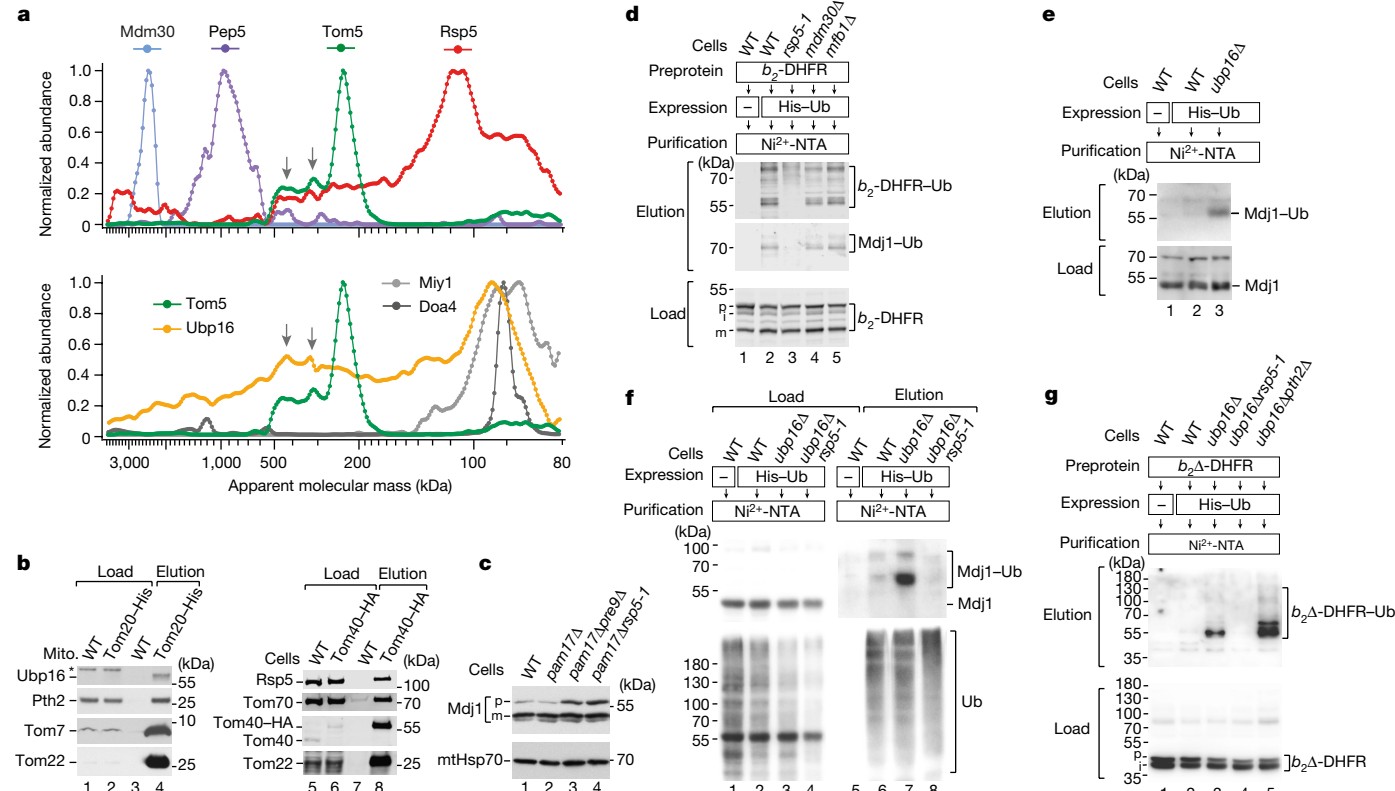

**Fig. 4 | Rsp5 and Ubp16 control ubiquitylation of mitochondrial precursor proteins. a**, Normalized abundance–mass profiles of the TOM subunit Tom5, the E3 ubiquitin ligases Mdm30, Pep5 and Rsp5 (top), and the deubiquitylating enzymes Ubp16, Doa4 and Miy1 (bottom). The arrows depict profile peaks of co-migration of Tom5, Rsp5 and Upb16. **b**, WT and Tom20–His mitochondria (left) or WT and Tom40–HA cell extracts (right) were lysed with digitonin and affinity-purified through Ni-NTA agarose or anti-HA affinity matrix. Proteins were analysed using SDS–PAGE and immunodetection. Load, 0.5%; elution, 100%. The asterisk marks an unspecific signal of anti-Ubp16. **c**, Cell extracts of the indicated strains were analysed by immunodetection. Load, 0.2%. p, precursor; m, mature form of Mdj1. Pre9, proteasomal subunit. **d**, WT, *rsp5-1*, *mdm30Δ* and

*mfb1Δ* cells expressing cytochrome $b_2$-DHFR and His-tagged ubiquitin as indicated were lysed under denaturing conditions and affinity-purified through Ni-NTA agarose. Proteins were analysed using SDS–PAGE and immunodetection. Load, 0.2%; elution, 100%. **e,f**, WT and *ubp16Δ* (**e**) and WT, *ubp16Δ* and *ubp16Δ rsp5-1* (**f**) strains expressing His-tagged ubiquitin were lysed under denaturing conditions and affinity-purified through Ni-NTA agarose. Proteins were analysed using SDS–PAGE and immunodetection. Load, 0.2%; elution, 100%. **g**, WT, *ubp16Δ*, *ubp16Δ rsp5-1* and *ubp16Δpth2Δ* cells expressing cytochrome $b_2$Δ-DHFR and His-tagged ubiquitin as indicated were lysed under denaturing conditions and affinity-purified through Ni-NTA. Proteins were analysed using SDS–PAGE and immunodetection. Load, 0.2%; elution, 100%.

preproteins from TOM[17]. A yeast double mutant lacking both *PTH2* and *UBX2* showed a synthetic growth defect and accumulation of Mdj1 precursor (Extended Data Fig. 10d,e), and tagged Ubx2 co-purified with TOM subunits and Pth2[17] (Fig. 5a). To test whether Pth2 cooperates with Ubx2 in TOM binding, we analysed the association of Ubx2 and Pth2 with the TOM complex in the reciprocal single-deletion mutants. The affinity purifications showed that the TOM–Ubx2 interaction does not depend on the presence of Pth2, and the TOM–Pth2 interaction does not depend on Ubx2; moreover, recruitment of Cdc48 to the mitochondria and the TOM complex occurred independently of Pth2 (Fig. 5b,c and Extended Data Fig. 10). Thus, Pth2 and Ubx2–Cdc48 bind to the TOM complex independently of each other.

Second, Vms1 is a ribosome-binding cytosolic peptidyl-tRNA hydrolase that facilitates the release and diminishes the aggregation of ribosome-stalled polypeptides during co-translational mitochondrial import[36–38]. A double deletion of *VMS1* and *PTH2* led to a strong synthetic growth defect (Extended Data Fig. 10d). The expression of a Pth2 mutant form with an inactive peptidyl-tRNA hydrolase (Pth2(D174A))[24,37] did not rescue the growth of *pth2Δvms1Δ* cells; however, a Pth2 mutant lacking the single mitochondrial outer-membrane anchor at the amino terminus (Pth2(ΔTM)) was able to rescue *pth2Δvms1Δ* cells (Fig. 5d,e). The mitochondrial localization of Pth2 is therefore not needed for its function as a peptidyl-tRNA hydrolase in the Vms1 pathway. By contrast, Pth2(D174A) but not Pth2(ΔTM) rescued the growth

of *pth2Δpam17Δ* cells (Fig. 5d), revealing that the role of Pth2 in the removal of accumulated preproteins requires its anchoring in the outer membrane. Thus, Pth2 functions in two different experimentally separable quality-control processes. Its peptidyl-tRNA hydrolase activity is required for Vms1-linked quality control at ribosome-stalled polypeptides, whereas the mitochondrial localization of Pth2 is needed for the removal of non-imported preproteins.

Third, Pth2 interacts with the nuclear and cytosolic ubiquitin-binding protein Dsk2 (dominant suppressor of *kar1*), a ubiquitin-like-domain-containing protein that shuttles ubiquitylated substrates to the proteasome[24,39]. The human homologues of Dsk2, ubiquilins, can bind to proteins with transmembrane segments, including non-imported mitochondrial outer membrane precursors, in the cytosol and deliver them for proteasomal degradation[40]. The identification of the TOM-linked Pth2 pathway raised the possibility that Dsk2 may participate in mitochondria-located quality control. Mutant cells lacking Dsk2 indeed displayed growth defects under respiratory conditions, particularly at elevated temperatures when high mitochondrial activity is required (Fig. 5f). The precursor form of Mdj1 accumulated in *dsk2Δpam17Δ* cells (Fig. 5g). Furthermore, increased amounts of ubiquitylated preproteins were detected in the absence of Pth2, the mitochondrial receptor of Dsk2[24] (Figs. 3e and 4g), indicating that Dsk2 is involved in the removal of non-imported ubiquitylated mitochondrial preproteins.

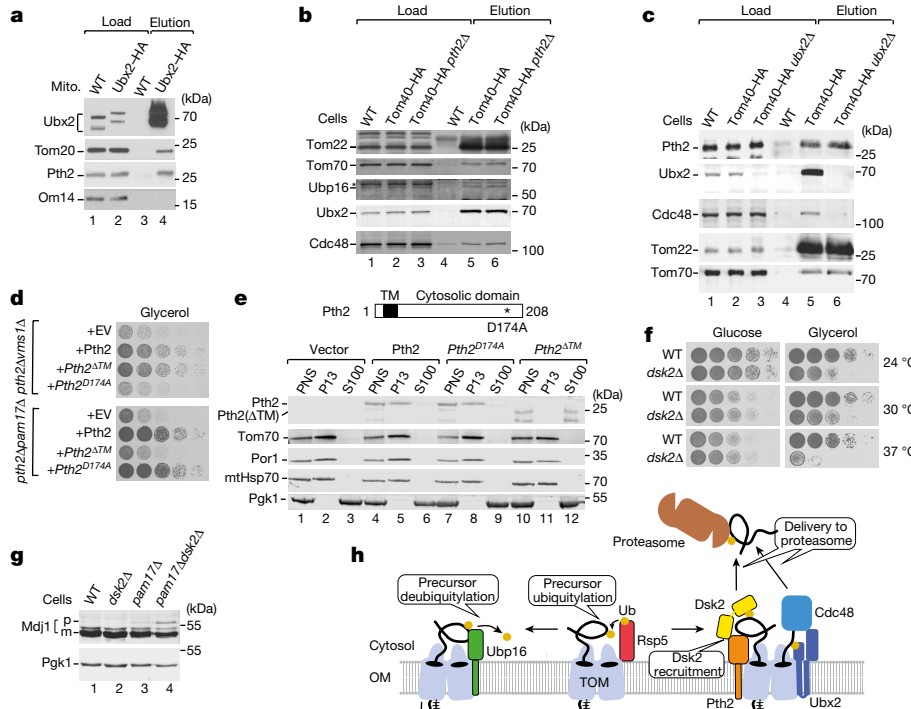

**Fig. 5 | The relationship of Pth2 to the quality-control factors Ubx2, Vms1 and Dsk2. a**, WT and Ubx2–HA mitochondria were lysed with digitonin and affinity-purified. Proteins were analysed using SDS–PAGE and immunodetection. Load 1%, elution 100%. **b**,**c**, WT, Tom40–HA and Tom40–HA*pth2Δ* (**b**) and Tom40–HA*ubx2Δ* (**c**) cell extracts were lysed with digitonin and affinity-purified. Proteins were analysed using SDS–PAGE and immunodetection. Load, 0.2%; elution, 100%. The asterisk marks an unspecific signal of anti-Ubp16. **d**, Serial dilutions of the indicated strains were spotted onto selective medium with glycerol as a carbon source and grown at 37 °C. EV, empty vector. **e**, Cartoon of the linear structure of Pth2. TM domain, transmembrane domain (amino acids 12–32). The amino acid exchange in the Pth2-mutant form is indicated. Cellular fractionation of the indicated strains. Post-nuclear supernatant (PNS) fractions enriched for mitochondria (P13) and the cytosol (S100) were analysed by immunodetection. Phosphoglycerate kinase 1 (Pgk1) was the cytosolic control. **f**, Serial dilutions of the indicated strains were spotted onto full medium with either glucose or glycerol as a carbon source. **g**, Cell extracts of the indicated strains were analysed by immunodetection. p, precursor; m, mature form of Mdj1. **h**, The proposed model of constitutive quality control for the removal of precursor proteins accumulated at the mitochondrial protein entry gate. Rsp5 ubiquitylates precursor proteins accumulated at the TOM complex. Ubp16 removes faulty ubiquitins from precursor proteins to enable their transport into the mitochondria. Ubx2 interacts with the TOM complex and cooperates with the AAA-ATPase Cdc48 in the transfer of ubiquitylated precursor proteins to the proteasome[17]. Pth2 binds to the TOM complex independently of Ubx2 and cooperates with Dsk2 in the transfer of precursor proteins to the proteasome. OM, outer membrane.

Taken together, the MitCOM analysis of TOM interactors uncovered factors for ubiquitylation/deubiquitylation at the mitochondrial import site and a Pth2–Dsk2 pathway for the removal of accumulated preproteins (Fig. 5h).

## Conclusions

MitCOM is a comprehensive high-resolution complexome of mitochondria. It covers more than 900 high-confidence yeast mitochondrial proteins and identifies more than 5,200 distinct protein peaks, revealing a considerable complexity of mitochondrial protein assemblies ranging from stable complexes to supercomplexes, assembly intermediates and dynamic interactors. The exquisite resolution and quantitative nature of the complexome enable a precise comparison of the shapes of co-migrating proteins and therefore efficient identification of protein interactors that are subsequently verified by biochemical and functional assays. The vast majority of mitochondrial proteins display multiple peaks in their abundance–mass profiles, with an average of six peaks per protein, indicating that interactions with more than one partner is a widespread characteristic. Owing to the stringency of our evaluation procedure, the peak numbers reported here probably underestimate the complexity of mitochondrial assemblies.

We used mitochondrial membrane protein complexes and quality control to illustrate the power of MitCOM for revealing interactions and pathways. (1) We identified the direct link of the Om14 receptor for cytosolic ribosomes[15] to the mitochondrial protein import site by identifying two different pools of Om14 in the mitochondrial outer membrane, one of them directly interacting with the TOM complex. (2) We identified the yeast J-protein partner Mdj2 of the prohibitin scaffold–AAA protease megacomplex of the mitochondrial inner membrane. (3) The identification of the peptidyl-tRNA hydrolase Pth2 at the TOM complex led us to identifying a quality-control pathway that interacts with TOM independently of the established mitoTAD pathway[17]. Together with a targeted screening for E3 ubiquitin ligases and deubiquitylases, this led to the identification of quality-control mechanisms that operate at the TOM complex (Fig. 5h). First, the multifunctional cytosolic E3 ligase Rsp5[29,31] ubiquitylates mitochondrial preproteins to facilitate their degradation. Second, Ubp16—a predicted outer-membrane deubiquitylase of previously unknown function—modulates preprotein quality control by removing ubiquitin moieties. Third, Pth2 constitutes an alternative degradation pathway by cooperating with ubiquilins[24,40] in the removal of accumulated preproteins. Thus, under constitutive conditions, protein entry into the mitochondria is controlled by coordinated ubiquitylation and deubiquitylation and at least two distinct transfer modes of non-imported preproteins to the proteasome for degradation. The quality-control factors are considerably less abundant than TOM[6] (Extended Data Fig. 9a), indicating that they transiently interact with the translocase. This underscores the high sensitivity of

the MitCOM approach in identifying dynamic protein interactions of low abundance.

A majority of processes and mechanisms of mitochondrial biogenesis, function and quality control have been conserved in evolution from yeast to humans, including numerous proteins that are linked to mitochondrial diseases[1,3,20]. The large number of assemblies and interactions resolved by MitCOM suggests that many mitochondrial functions and pathways await their assignment, for physiological as well as pathophysiological processes.

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

## Methods

### Yeast strains and growth conditions

The yeast strains used in this study are derived from *S. cerevisiae* BY4741 and YPH499 (Supplementary Table 4). The wild-type strain YPH499 as well as the mutant strains $Cox4_{His}$, $Tom20_{His}$, $Tom40_{HA}$, $Tom40_{HA}ubx2\Delta$, $ubx2\Delta$, $pam17\Delta$ and $pre9\Delta pam17\Delta$ were described previously[17,42,43]. The wild-type strain BY4741 as well as the mutant strains $pth2\Delta$, $ubx2\Delta$, $ubp16\Delta$, $rsp5$-$1$, $mdm30\Delta$, $mfb1\Delta$ and $vms1\Delta$ were obtained from EUROSCARF. Tagging and deletion of open reading frames were performed by homologous recombination using DNA cassettes amplified by PCR using Taq and Vent polymerase (NEB)[44] or KOD hot-start DNA-Polymerase (Merck Millipore). The genetic information for a triple HA tag was chromosomally introduced before the STOP codon of the open reading frame of *TOM40* in $pth2\Delta$ and $tom70\Delta$ cells using the cassette amplified from pFA6a 3HA-His3MX6[45]. The open reading frame of *DSK2* was deleted using the pFA6a KanMX4 cassette[46]. Deletion of *PTH2* in $ubp16\Delta$ and $ubx2\Delta$, *VMS1* in $pth2\Delta$ and *UBP16* in $rsp5$-$1$ was performed using the pFA6A His3MX6 cassette[46]. Deletion of *PAM17* in the $pth2\Delta$, $dsk2\Delta$, $rsp5$-$1$, $ubp16\Delta$, $mfb1\Delta$, $mdm30\Delta$, $hrd1\Delta$ and $doa10\Delta$ strains, and deletion of *TOM70* in the BY4741 and $Tom40_{HA}$ strains was performed using the pFA6a hphNT1 cassette[44] (the deletion cassettes used are shown in Supplementary Table 5). Yeast strains were cultured according to standard protocols at temperatures between 24 °C and 37 °C in complete YP-medium (1% (w/v) yeast extract, 2% (w/v) bacto-peptone) or selective minimal medium (SM) (0.67% (w/v) yeast nitrogen base with ammonium sulfate; 0.07% (w/v) amino acid mixture) containing 2% (w/v) glucose (YPD, SMD), 2% (w/v) sucrose (YPS, SMS), 2% (w/v) galactose (YPGal, SMGal) or 3% (w/v) glycerol as carbon source. The cell cultures were grown until the early logarithmic growth phase, on the basis of the optical density at a wavelength of 600 nm ($OD_{600}$).

### Construction of plasmids

Pth2 was cloned with its endogenous promoter (962 bp upstream of the start codon) into pRS416[43]. The Pth2 D174A and $b_2$-DHFR$^{GGxY}$ L251G/P252G mutations were generated by site-directed mutagenesis. The putative transmembrane domain of Pth2 (amino acids 12–32) was deleted by PCR amplification of the entire plasmid without the region encoding amino acids 12–32 followed by in vitro recombination using HiFi assembly (NEB). A list of the plasmids used is provided in Supplementary Table 5. Plasmids were used for the expression of cytochrome $b_2$ precursor variants containing DHFR. Folding of the DHFR domain prevents the complete translocation of the preproteins through the TOM channel and therefore arrests the N-terminal $b_2$-part in the mitochondrial import site. Two types of $b_2$-DHFR precursors were accumulated in the import site—$b_2\Delta$-DHFR, which carries a matrix-targeting signal, and $b_2$-DHFR, which contains both a matrix-targeting signal and an inner membrane sorting signal[17,47].

### Growth analysis of yeast strains

For comparing the growth of different yeast strains, exponentially growing cells were diluted to an $OD_{600}$ of 1.0 and diluted 1:5 (five times). The dilutions were spotted onto YP or SM plates containing glucose or glycerol as the sole carbon source and incubated at the indicated temperatures. Pictures of plates were taken after 1 to 4 days, depending on growth temperature and carbon source.

### Isolation of mitochondria

Purification of mitochondria was performed by differential centrifugation[48]. Yeast cells were collected at an early logarithmic growth phase (5,500$g$, 8 min, 24 °C). Cells were washed with distilled $H_2O$ and resuspended in DTT buffer (100 mM Tris/HCl pH 9.4, 10 mM dithiothreitol (DTT)) at a concentration of 2 ml per g wet weight of the cell pellet, followed by incubation for 30–45 min at growth temperature under constant shaking. Cells were then washed in zymolyase buffer (1.2 M sorbitol, 20 mM KP$_i$ pH 7,4) and resuspended in zymolyase buffer at a concentration of 7 ml per g of cells. Subsequently, cells were incubated with 4 mg zymolyase per g of cells under constant shaking for 30–45 min at growth temperature to digest the cell wall. Next, cells were pelleted (2,500$g$, 5 min, 24 °C) and washed once with zymolyase buffer. The obtained spheroplasts were resuspended in ice-cold homogenization buffer (0.6 M sorbitol, 10 mM Tris/HCl pH 7.4, 1 mM ethylenediaminetetraacetic acid (EDTA), 1 mM phenylmethylsulfonylfluoride (PMSF), 0.2% (w/v) bovine serum albumin) using 6.5 ml of buffer per g of cells. Cells were homogenized using a glass potter with 15 strokes up and down. Subsequently, cell debris and large organelles like the nucleus were removed (2,500$g$, 5 min, 4 °C). The supernatant was centrifuged to isolate the mitochondria (17,000$g$, 15 min, 4 °C). The mitochondrial pellet was resuspended in SEM buffer (250 mM sucrose, 10 mM MOPS/KOH pH 7.2, 1 mM EDTA) and washed again in SEM buffer. The isolated mitochondria were resuspended in SEM buffer. The protein concentration was determined using the Bradford assay and mitochondria were aliquoted at a protein concentration of 10 mg ml$^{-1}$. Mitochondria were frozen in liquid nitrogen and stored at −80 °C.

### Cryo-slicing blue native gel electrophoresis

For high-resolution complexome profiling, a blue native gradient gel (2–13% (w/v) acrylamide, 0.06–0.40% (w/v) bis-acrylamide, 67 mM ε-amino *n*-caproic acid, 50 mM Bis-Tris/HCl, pH 7.0) was used. Mitochondria corresponding to 1 mg protein amount were pelleted and solubilized in 0.8 ml lysis buffer (20 mM Tris/HCl pH 7.4, 0.1 mM EDTA, 50 mM NaCl, 10% (v/v) glycerol) containing 1% (w/v) digitonin for 30 min on ice (we used optimized conditions for a mild and efficient lysis of the yeast mitochondrial preparation by applying 8 mg purified digitonin to 1 mg mitochondrial protein; the protein to digitonin ratio of 1:8 enables efficient extraction of yeast mitochondrial membrane protein complexes, yet is milder than the 1:10 protein:digitonin ratio that is often used for yeast mitochondria[49] or the application of other detergents[8]; this is illustrated by the predominant presence of the physiological, fully assembled dimer of the $F_1F_O$-ATP synthase in comparison to the monomer (Fig. 1b and Extended Data Fig. 8d), whereas the typical 1:10 ratio conditions lead to an about equal distribution between dimer and monomer on blue native gels[49]). Subsequently, the sample was loaded onto a sucrose gradient consisting of 50% (w/v) sucrose and 20% (w/v) sucrose. After centrifugation, the upper phase was removed and the remaining supernatant was mixed with loading dye (0.5% (w/v) Coomassie G-250, 50 mM ε-amino *n*-caproic acid, 10 mM Bis-Tris/HCl, pH 7.0). The sample was applied to a loading zone of 5 cm width. Electrophoresis was performed at 15 mA in the presence of BN cathode buffer (0.02% (w/v) Coomassie G-250, 50 mM Tricine, 15 mM Bis-Tris/HCl, pH 7.0) and anode buffer (50 mM Bis-Tris/HCl, pH 7.0). After 1 h, the BN cathode buffer was replaced by a cathode buffer lacking Coomassie G-250 and the electrophoresis was continued for 2.5 h at 15 mA. A 2.5 cm lane was then excised, fixed in 30% ethanol/15% acetic acid, embedded in tissue embedding medium (Leica) and subjected to cryo-slicing[50]. Using a step size of 0.3 mm along the gel lane, 245 slices were obtained, extensively washed and separately digested with trypsin[50].

### MS analysis

The trypsin-digested peptides were dissolved in 20 μl sample buffer (0.5% (v/v) trifluoroacetic acid in $H_2O$) and 1 μl aliquots (or less) were taken for LC–MS/MS analysis. Loading onto the precolumn (PepMap 100, C18 stationary phase) was achieved through an autosampler of a split-free UltiMate 3000 RSLCnano HPLC (Dionex/Thermo Fisher Scientific). Subsequent elution and separation on the SilicaTip column emitter (inner diameter, 75 μm; tip, 8 μm; New Objective,; packed 23 cm with ReproSil-Pur 120 ODS-3 (C18 stationary phase; Dr. Maisch HPLC)) occurred during a three-step linear gradient generated from eluent A (0.5% (v/v) acetic acid) and eluent B (0.5% (v/v) acetic acid in 80% (v/v) acetonitrile): after 5 min equilibration in 3% B, 90 min from

3% B to 30% B; 20 min from 30% B to 50% B; and 10 min from 50% B to 99% B. Subsequent column washing/regeneration comprised 5 min 99% B; 5 min from 99% B to 3% B; 10 min 3% B. The flow rate was set to 300 nl min⁻¹. Electrospray parameters were positive ion mode, spray voltage 2.3 kV, transfer capillary temperature 300 °C. Data were acquired on the QExactive HF-X mass spectrometer (Thermo Fisher Scientific) with the following settings: maximum MS/MS injection time = 200 ms; dynamic exclusion time = 45 s; minimum signal intensity threshold = 40,000 (counts), fragmentation = 15 top precursors; mass isolation width = 1.0 $m/z$.

### Protein identification

Primary MS data were processed using msconvert (https://proteowiz-ard.sourceforge.io; v.3.0.11098; settings: Mascot generic format, filter options 'peakPicking true 1', 'threshold count 500 most-intense'). Obtained peak lists were then $m/z$-calibrated using the MaxQuant raw file processor (v.1.6.17, https://www.maxquant.org)[51,52] and used as input for a Mascot (v.2.7, Matrix Science) database search against the UniProtKB/SwissProt yeast database (SwissProt_YEAST_20201007) with general contaminations added (GPM cRAP database; cRAP_20190304). Match parameters were as follows: precursor mass tolerance = ±2.5 ppm, variable modifications = acetyl (protein N-term), carbamidomethyl (C), formyl (N-term, S, T), Gln->pyro-Glu (N-term Q), Glu->pyro-Glu (N-term E) and oxidation (M), fragment mass tolerance = ± 20 mmu, missed tryptic cleavage(s) = 1. Export filter settings were as follows: peptide-spectrum-match (PSM) FDR = 3%, minimum ion score = 0.5, grouping of related protein hits used the name of the predominant member. Exogenous contaminants (for example, keratins, trypsin, IgG chains) or protein identifications based on only one specific peptide in less than three slice samples were not considered further.

### Protein quantification

For label-free quantification of proteins, LC–MS data were processed as previously described[9,50,53,54] with some crucial improvements. Max-Quant (v.1.6.17; https://www.maxquant.org)[51,52] was used to determine and mass-calibrate peptide signal intensities (peak volumes) from recorded FT full scans. Systematic variations in peptide elution times were corrected by LOESS regression after pairwise alignment of the datasets (median peptide elution times over all of the aligned data-sets were used as a reference). Matching of peak volumes and peptide identities (obtained either directly or indirectly from MS/MS database matches) was achieved using custom developed software[53] ($m/z$ and elution time matching tolerances were ±1.5–2 ppm and ±1 min, respectively). Global offsets in peptide intensity between runs were corrected by normalization to the local median of the relative peptide intensities (consistently assigned peptides within a window of 40 slices). For effectively eliminating the impact of missing, non-consistent or incorrectly assigned peptide intensity values, we applied an additional procedure consisting of four steps. First, the accuracy of all assigned peptide intensity values was determined by analysing matrices (pep-tides versus runs) of protein-specific peptide intensity values for their internal consistency. Within these matrices, the pairwise relationships of peptide intensity values between and within MS-runs (in all possi-ble combinations) provided distributions of predicted intensities for connected matrix cells from which expected intensity values (EPVs) could be calculated (by kernel density estimation), which served as measure of consistency of the respective peak volume values and were later used as weighting factors. If insufficient data precluded determi-nation of EPVs, values interpolated from EPV-validated intensities in neighbouring (window of 5) slices/datasets were used alternatively. Second, a time- and run-dependent detectability threshold was esti-mated for each of the matrix cells (peptides versus runs) using the third percentile of intensity values from peptides co-eluting within a 3 min time window. Third, for each protein, peptide intensity values from qualified runs (that is, consistent protein-specific peak volume

values of respective columns in each protein matrix) were merged (EPV-weighted least squares fits to the dataset with the highest num-ber of peptide intensities assigned to the respective protein) into a single intensity value vector, termed protein reference ridge. These vectors reflected the maximum protein coverage of MS/MS-identified and quantified peptides with their relative ionization efficiencies and were used to determine molecular abundances (abundance_norm spec values) as described previously[53]. Fourth, protein quantification was achieved by a weighted fitting of its measured peptide intensities in five consecutive slices (equivalent to a sliding average) to its reference ridge (Extended Data Fig. 2). Quantification details for each data point in a protein profile (Supplementary Table 2) are provided in the 'peptide details' feature of the expert viewer tool (Extended Data Fig. 3; https://www.complexomics.org/datasets/mitcom).

The oversampling of the blue native gel separation (0.3 mm step size) was used to provide robust protein quantification without com-promising effective size resolution. Thus, each protein abundance value, that is, each data point in an abundance mass profile, integrated the results from five consecutive LC–MS/MS analyses. Moreover, the processing of LC–MS input data described above provided objective measures for reliability and accuracy of protein quantification: (1) the number of protein-specific peptide intensities used and (2) the devia-tion of these peptide intensities from the expected peptide intensities. This information was integrated into a 'reliability score' (ranging from 0 (not significant) to 1 (maximum reliability)) for each protein abun-dance value (Supplementary Table 2).

### Evaluation, accession and visualization of data

A total of 1,891 proteins with a required minimal number of two protein-specific peptides were identified in the entire csBN–MS dataset. Of these, 906 proteins were considered to be bona fide mito-chondrial proteins on the basis of respective annotations in either the UniProtKB/SwissProt database, the Yeast Genome Database (SGB) or ref. [6]. Together, the 906 mitochondrial proteins represent the core of the complexome dataset termed the MitCOM (Supplementary Tables 1 and 2). Information on (1) molecular mass and membrane-spanning helices was inferred from the UniProtKB/SwissProt database, (2) locali-zation in submitochondrial compartments was extracted from the respective SGD GO terms, and (3) functional classification was taken from ref. [6] supplemented by GO annotations and literature for the proteins not already classified therein (Fig. 2, Supplementary Table 1 and Extended Data Fig. 6). Among the proteins identified with one protein-specific peptide only, an additional 49 mitochondrial proteins were found to be potentially significant and were separately added to the list of MitCOM proteins in Supplementary Table 1. The remaining 985 proteins, all identified by at least two protein-specific peptides, were classified as non-mitochondrial proteins predominantly localized in the endoplasmic reticulum and cytosol (Fig. 1, Extended Data Fig. 7 and Supplementary Table 3). Information on their molecular mass and membrane integration, their preferred subcellular localization and their functional classification was obtained as described for the mitochondrial proteins above.

The principles of mass estimation of protein complexes using blue native electrophoresis were outlined in ref. [55]. For converting slice num-bers to apparent molecular masses of proteins/complexes we used a set of marker proteins/complexes with a defined migration pattern (that is, sharply focused profile peaks) and molecular masses broadly distributed over the sampled blue native gel range (name/subunits with references; predicted molecular mass, log[molecular mass], slice maxi-mum): oxoglutarate dehydrogenase/ketoglutarate dehydrogenase complex (Kgd1/Odo1–Kgd2/Odo2–Lpd1/Dldh–Kgd4[56–60]; 3,020 kDa, 3.48, 21.4); pre-60S ribosome large subunit (RL3, RLP24, NOG1; PDB: 3JCT (ref. [61]); 2,680 kDa, 3.43, 11.5); dimer of $F_1F_0$-ATP synthase (com-plex V dimer[8]; 1,250 kDa, 3.10, 54.1); respiratory $III_2IV_2$ supercomplex (ref. [8]; 1,000 kDa, 3.00, 62.4); respiratory $III_2IV_1$ supercomplex (ref. [8];

750 kDa, 2.88, 71.6); monomer of $F_1F_0$-ATP synthase (complex V monomer[8]; 600 kDa, 2.78, 85,6); SAM–Mdm10 complex (PDB: 7BTX (ref. [62]); 185.5 kDa, 2.27, 143.8); TIM22 complex (PDB: 6LO8 (ref. [63]); 174 kDa, 2.24, 136.2); ATM1 (ABC transporter mitochondrial; PDB: 4MYC (ref. [64]); 155 kDa, 2.19, 157); $SAM_{core}$ complex (Sam50, Sam37, Sam35[62,65]; 129.4 kDa, 2.11, 173.5); succinate dehydrogenase (complex II[8]; 130 kDa, 2.11, 182); NADP-cytochrome P450 reductase (Ncp1/NCPR[66]; 77 kDa, 1.88, 239). The resulting relationship of molecular mass over slice numbers was fitted with a sigmoidal function (IGOR Pro 9 WaveMetrics) and the resulting fit-line was used for calibration (Extended Data Fig. 2d).

Abundance–mass profiles of all MitCOM proteins (Supplementary Table 2) were analysed for their composition of individual components (peaks) using custom-developed software (complexomics-mitcom v.1.0; released as a Python package under the MIT license, available at Zenodo (https://doi.org/10.5281/zenodo.7355040)). First, locations of apparent peaks were determined by local maxima search with subsequent filtering (minimum relative height 0.1, minimum relative prominence 0.5, maximum width 50). Then, a multicomponent Gaussian model was initialized with the number and locations of the identified peaks. The model was iteratively adjusted and fitted to the profile. Up to 12 Gaussian components were added preferably at locations with large residuals, resulting in an improved fitting of highly overlapping peaks, manifesting as 'shoulders'. Sensible limits and stop conditions were applied to avoid overfitting. A total of 818 MitCOM proteins was accessible to automated analysis. From a total number of 5,224 peaks (manually curated), 4,070 peaks were adequately fitted by our algorithm providing parameters for their apparent mass, half-width and molecular abundance (Fig. 1c). The total peak count (manually curated) was used for statistical analysis in Fig. 2 and Extended Data Fig. 6. For the non-mitochondrial proteins, the number of apparent peaks was counted manually (Extended Data Fig. 7).

Abundance–mass profiles of all MitCOM proteins were deposited in the openly accessible interactive resource platform Complexome Profiling Data Resource (CEDAR)[67], where they can be accessed through an interactive online visualization tool (https://www3.cmbi.umcn.nl/cedar/browse/experiments/crx36). Protein profile normalization, filtering, baseline subtraction and magnification/scaling for convenient display and evaluation are integrated functions of this viewer (Extended Data Fig. 3). Moreover, using custom Pearson correlation analysis, it offers a basic method to search for proteins with similar abundance profile peaks or patterns. An extended version of the viewer with additional features enabling the inspection of the peptide intensity information underlying each datapoint of the protein profiles (Extended Data Fig. 3), an integrated help function explaining the use of the available features and a supplemental viewer containing abundance–mass profiles of the quantified non-mitochondrial proteins (that passed a quality check) are available online (https://www.complexomics.org/datasets/mitcom).

For a global view on the complexome organization of MitCOM (Extended Data Figs. 4 and 5), unsupervised protein profile matching was performed as follows (similar to the approach in ref. [11]). Successfully fitted protein profile components (Fig. 1c) were filtered by width (max 20 s.d.) and location (peak full width at half maximum fully inside gel slice range) and clipped at 2.5 s.d. These defined the boundaries of 3,263 profile segments as seed regions of interest (ROIs). Comparison of each ROI to its corresponding segment in other profiles using Pearson correlation yielded 82,268 high-correlating segments ($r \geq 0.95$) as additional ROIs. Finally, out of the total of 85,531 ROIs, the ones that originated from the same profile and had similar boundaries (within 3 slices) were merged. The resulting 49,112 ROIs were used to assess similarity of protein components. To this end, a distance metric was designed that incorporated the following ROI-specific values: (1) slice index of left boundary; (2) slice index of right boundary; (3) maximum abundance; (4) $r$ value from correlation with seed ROI (average if ROIs were merged). All of the values were minimum/maximum-normalized, except for boundaries, which were square-root-transformed, giving the highest weight to component locations and quickly penalizing differences. The distance of any two ROIs is determined by the Euclidean distance of their respective value vectors and the coefficient $r$ of their mutual correlation. With the maximum theoretical distance being a dataset-specific fixed value, distances could easily be converted to a normalized similarity score ranging from 0 (most dissimilar) to 1 (identical location and abundance values). On the basis of the custom distance metric, pairwise distances of all protein ROIs were calculated and used for visualizing component similarity using a $t$-SNE plot (Extended Data Fig. 5).

## Preparation of yeast cell extracts

Whole yeast cell lysates were obtained by post-alkaline extraction[68]. Exponentially growing yeast cells ($OD_{600}$ of 2.5) were pelleted (3,000$g$, 5 min, 20 °C) and resuspended in distilled $H_2O$. Subsequently, the samples were mixed with the same volume of 0.2 M NaOH and incubated for 5 min at 24 °C. Cells were pelleted (3,000$g$, 5 min, 4 °C), resuspended in sample buffer (8 M urea, 5% (w/v) SDS, 1 mM EDTA, 1.5% (w/v) DTT, 0.025% (w/v) bromophenol blue, 200 mM Tris/HCl pH 6.8) and denatured for 10 min at 65 °C shaking at 1,400 rpm.

Yeast extracts for large-scale affinity purification were generated by collecting cells in the early logarithmic growth phase (5,500$g$, 8 min, 24 °C). Cells were then washed with distilled $H_2O$ lysis buffer (0.1 M EDTA, 50 mM NaCl, 10% (v/v) glycerol, 20 mM Tris/HCl pH 7.4). Cells resuspended in lysis buffer were then frozen in liquid nitrogen and cell disruption was achieved by cryo-grinding at 25 Hz for 10 min in a Cryo Mill (Retsch). The obtained lysates were stored at −80 °C until further use.

To obtain small amounts of cell extracts for affinity purifications, cells from exponentially growing cells ($OD_{600}$ of 100–200) were resuspended in lysis buffer with protease inhibitors (1 mM PMSF, 1× HALT protease inhibitor cocktail (Thermo Fisher Scientific)). Cells were then ruptured with silica beads 6 times 30 s with a 1 min break in between at 4 °C on a cell disruptor (Vortex Disruptor Genie). Cell extracts were cleared by centrifugation (2,000$g$, 5 min, 4 °C) and directly further processed for affinity purification.

## Affinity purification of tagged proteins

For affinity purification of His-tagged proteins[17,42,69], isolated mitochondria were solubilized in lysis buffer (20 mM Tris pH 7.4, 0.1 mM EDTA, 50 mM NaCl, 10% (v/v) glycerol, 2 mM PMSF, 1× protease inhibitor cocktail without EDTA) containing 1% (w/v) digitonin and 10 mM imidazole and incubated for 15 min at 4 °C. The solubilized sample was cleared by centrifugation (10 min, 17,000$g$, 4 °C) and incubated with Ni-NTA agarose beads (Qiagen) that were pre-equilibrated in lysis buffer with 0.1% (w/v) digitonin and 10 mM imidazole. After 1 h incubation at 4 °C under constant rotation, unbound proteins were removed and the affinity matrix was washed with excess amount of lysis buffer containing 0.1% (w/v) digitonin and 40 mM imidazole. Bound proteins were eluted with lysis buffer containing 0.1% (w/v) digitonin and 250 mM imidazole. After addition of sample buffer proteins were denatured for 10 min at 60 °C (Cox4–His) or 5 min at 96 °C.

For affinity purification of HA-tagged proteins[17,69], isolated mitochondria or cell extracts were solubilized in lysis buffer containing 1% (w/v) digitonin. After solubilization for 15 min (purified mitochondria) or 30 min (cell extracts), the samples were cleared by centrifugation (17,000$g$, 10 min, 4 °C). The supernatant was incubated for 1 h at 4 °C under constant rotation with an anti-HA affinity matrix (Roche) that was pre-equilibrated with 0.5 M acetate followed by washing with lysis buffer containing 0.1% (w/v) digitonin. Unbound proteins were removed and the beads were washed with an excess amount of lysis buffer containing 0.1% (w/v) digitonin. Proteins were eluted by incubation with sample buffer at 95 °C.

For purification of Phb1–protein A, mitochondria were solubilized in lysis buffer containing 1% (w/v) digitonin for 1 h at 4 °C. The lysate

was cleared by centrifugation (17,000$g$, 10 min, 4 °C) and incubated for 1.5 h at 4 °C with IgG Sepharose (Cytiva). The IgG Sepharose was washed 10 times with wash buffer (20 mM Tris-HCl pH 7.4, 60 mM NaCl, 10% (v/v) glycerol, 0.1 mM EDTA, 2 mM PMSF, 0.3% (w/v) digitonin). Phb1 and bound proteins were eluted by cleavage of the protein A tag using TEV protease (Thermo Fisher Scientific) in wash buffer at 24 °C for 2.5 h. Subsequently, the protease was removed through its His tag by incubating the eluates with equilibrated Ni-NTA for 30 min at 4 °C.

For tandem purification of the TOM complex[70,71] through Tom22–His and Tom40–Strep, isolated mitochondria were solubilized in tandem buffer (100 mM Tris/HCl pH 8.0, 150 mM NaCl, 10% (v/v) glycerol, 1× protease inhibitor cocktail) supplemented with 5 mM imidazole and 3% (w/v) digitonin for 1 h rotating. The samples were cleared (17,000$g$, 10 min, 4 °C) and the subsequent purification steps were performed using the ÄKTA Explorer 100 system (Cytiva). In the first steps, Tom22–His was purified using the HisTrap HP column pre-equilibrated with tandem buffer containing 5 mM imidazole and 0.1% (w/v) digitonin. The column was washed with tandem buffer containing 5 mM imidazole and 0.1% (w/v) digitonin and bound proteins were eluted with tandem buffer containing 250 mM imidazole and 0.1% (w/v) digitonin. In the second purification step via Tom40–Strep, the elution sample was applied onto a Strep-Tactin HP column pre-equilibrated with tandem buffer containing 0.1% (w/v) digitonin. The column was washed with excess amount of tandem buffer containing 0.1% (w/v) digitonin and bound protein was eluted with tandem buffer containing 0.1% (w/v) digitonin and 10 mM biotin.

### Purification of ubiquitin-modified proteins

Proteins conjugated to His-tagged ubiquitin were purified through Ni-NTA agarose under denaturing conditions[41,72]. Cells expressing His-tagged ubiquitin were grown to logarithmic growth phase. Cells corresponding to an OD$_{600}$ of 200 were collected (2,500$g$, 4 min, 4 °C) and washed with distilled $H_2O$. Cells were resuspended under denaturing conditions to inactivate proteases including deubiquitylating enzymes in 1 ml buffer A (6 M guanidinium hydrochloride, 100 mM NaH$_2$PO$_4$, 10 mM Tris-HCl, pH 8.0). Silica beads (diameter 0.5 mm) were added to the samples and cells were disrupted using a cell disruptor (Vortex Disruptor Genie). Cellular lysates were cleared (500$g$, 5 min, 4 °C) to remove residual beads. Subsequently, the samples were diluted 1:10 in the presence of 0.05% (v/v) Tween-20 and incubated for 1 h at 24 °C under constant rotation. Insoluble material was removed (3,500$g$; 10 min; 4 °C) and the remaining supernatants were incubated in the presence of 20 mM imidazole with Ni-NTA agarose beads overnight at 4 °C under constant rotation. After removal of unbound proteins, the beads were washed twice with buffer A containing 20 mM imidazole and 0.05% Tween-20, followed by five washing steps with buffer C (8 M urea, 100 mM NaH$_2$PO$_4$, 10 mM Tris-HCl, pH 6.3, 0.05% Tween-20, 20 mM imidazole). Bound proteins were eluted with HU sample buffer (8 M urea, 5% (w/v) SDS, 1 mM EDTA, 1.5% (w/v) DTT, 0.025% (w/v) bromophenol blue, 200 mM Tris-HCl pH 6.8) for 10 min at 65 °C.

### Subcellular fractionation

Cells were fractionated by differential centrifugation[17]. Cells were grown to an early logarithmic growth phase. Cells corresponding to an OD$_{600}$ of 100 were resuspended in DTT buffer (100 mM Tris pH 9.4, 10 mM DTT) and incubated at 30 °C (20 min, 900 rpm). Cells were then resuspended in zymolyase buffer (1.2 M sorbitol, 20 mM KP$_i$ pH 7.4), zymolyase was added at a final concentration of 100 mg ml$^{-1}$ and cells were incubated at 30 °C (45 min, 900 rpm). Cells were washed with zymolyase buffer and resuspended in ice cold homogenization buffer (0.6 M sorbitol, 10 mM Tris/HCl pH 7.4, 1 mM EDTA, 1 mM PMSF, 0.2% (w/v) bovine serum albumin). Cells were homogenized using a glass potter with 20 strokes up and down. Subsequently, cell debris and large organelles such as the nucleus were removed (1,500$g$, 5 min, 4 °C). A fraction of the supernatant was collected as post-nuclear supernatant,

the remaining supernatant was subjected to centrifugation to isolate mitochondria (17,000$g$, 15 min, 4 °C). The mitochondrial pellet was resuspended in SEM buffer (250 mM sucrose; 10 mM MOPS/KOH pH 7.2; 1 mM EDTA), mitochondria were purified twice by centrifugation through a sucrose cushion (500 mM sucrose; 10 mM MOPS/KOH pH 7.2; 1 mM EDTA) and mitochondria were resuspended in SEM buffer (P13 fraction). The supernatant of the first 17,000$g$ centrifugation was ultracentrifuged (100,000$g$, 1 h, 4 °C) and the supernatant was collected as the S100 fraction.

### Blue native gel electrophoresis for standard analysis

For blue native gel separation[48], mitochondria were solubilized in lysis buffer (20 mM Tris/HCl pH 7.4, 0.1 mM EDTA, 50 mM NaCl, 10% (v/v) glycerol) containing 1% (w/v) digitonin. The samples were cleared by centrifugation (17,000$g$, 15 min, 4 °C) and loading dye was added to a final concentration of 0.5% (w/v) Coomassie Brilliant Blue G-250, 10 mM Bis-Tris pH 7.0 and 50 mM 6-aminocaproic acid before loading onto the blue native gel. The prime symbols at the molecular mass markers of the blue native gels in Fig. 3b and Extended Data Fig. 10b,h indicate the correlation between the migration of water-soluble markers and membrane-protein markers according to ref. [62].

### Immunoblotting

Proteins separated on polyacrylamide gels were transferred by semi-dry blotting to a polyvinylidene fluoride membrane (Milipore) with blotting buffer (20% (v/v) methanol, 150 mM glycine, 0.02% (w/v) SDS, 20 mM Tris base) for 2 h at 250 mA. After blotting, membranes were blocked with 5% (w/v) skimmed milk powder in TBS-T (12.5 mM NaCl, 20 mM Tris/HCl, pH 7.4, 0.1% (v/v) Tween-20) or RotiBlock (Roth) for 1 h. The membranes were incubated with primary antibodies for 1–2 h at room temperature or overnight at 4 °C. The membranes were washed with an excess of TBS-T and incubated with secondary antibodies at room temperature for 1 h. Different types of secondary antibodies were used. For detection of immune signals using the Licor system, secondary antibodies against rabbit or mouse were coupled to fluorescent labels (IRDye 800CW, anti-mouse; IRDye 800CW, anti-rabbit; IRDye 680RD, anti-mouse). For detection with the image analyser or X-ray films, an anti-rabbit antibody coupled to horseradish peroxidase was used. Unbound antibodies were removed by washing with excess of TBS-T. The signal of horseradish peroxidase coupled secondary antibodies was detected after incubating the membrane with enhanced chemiluminescence solution[73] using either an Amersham Imager 680 (Cytiva) or the LAS3000 image reader (FujiFilm). Fluorescent secondary antibodies were detected on Odyssey CLx Infrared Imaging System (Li-Cor) and analysed using Image Studio (v.5.2.5; Li-Cor). The specificity of the immunosignals was confirmed by their absence in cells or mitochondria from deletion strains. In case of essential genes, the size shift of the band in cells expressing tagged-proteins confirmed the specificity of the immunosignal. The separating white lanes indicate where irrelevant gel lanes were digitally removed. A list of the antibodies used is provided in Supplementary Table 6. Experiments were typically run on several gels, which were analysed in parallel by western blotting, including control western blots. Representative blots were selected and processed from the original files (Supplementary Fig. 1) using ImageJ (v.2.1.0), Adobe Photoshop 2021 and Adobe Illustrator 2021.

### Reproducibility and image processing

Representative images are shown for growth and biochemical assays/western blotting, including analysis of yeast growth (wild-type and mutants), total cell extracts, affinity purification from cell extracts, subcellular fractionation, protein steady state levels, blue native electrophoresis and affinity purification from isolated mitochondria. The findings were confirmed by independent experiments for the following figures (minimum number of independent experiments in parentheses): Figs. 3b (2), 3d (3), 3e (2), 4b (2), 4c (2), 4d (2), 4e (2), 4f (2), 4g (2),

5a (2), 5b (3), 5c (2), 5d (3), 5e (2), 5f (3) and 5g (2) and Extended Data Figs. 8f (3), 8h (2), 9b (2), 9c (2), 9d (2), 9e (2), 9f (5), 9g (3), 9h (2), 10a (2), 10b (2), 10c (3), 10d (2), 10e (2), 10f (2), 10g (2) and 10h (2). Images of western blots and growth assays were processed using Adobe Photoshop 2021 and figures were assembled in Adobe Illustrator 2021.

## Reporting summary

Further information on research design is available in the Nature Portfolio Reporting Summary linked to this article.

## Data availability

The MS data generated during this study are available at the ProteomeXchange under identifier PXD029548. UniProtKB/Swiss-Prot (https://www.uniprot.org; SwissProt_YEAST_20201007) and *Saccharomyces* Genome Database (SGD, https://www.yeastgenome.org; GO term mapping 20201027) were used as resources for structural, cell biological and functional annotation of proteins. The MitCOM datasets, including an interactive profile viewer, are available at Complexomics (https://www.complexomics.org/datasets/mitcom) or on the CEDAR platform (https://www3.cmbi.umcn.nl/cedar/browse/experiments/CRX36). All other data are available in the figures and the Supplementary Information, including uncropped versions of gels/blots in Supplementary Fig. 1 and Supplementary Tables 1–3 (Excel files).

## Code availability

Complexomics-mitcom v.1.0 (a Python package for peak detection, similarity score calculation and *t*-SNE visualization) is available at Zenodo (https://doi.org/10.5281/zenodo.7355040).

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

**Acknowledgements** We thank J. Jordan for help with bioinformatics; M. Glickman for providing the His-ubiquitin plasmid; and O. Neuber for discussions. Work included in this study has also been performed in partial fulfilment of the requirements for the doctoral theses of A.G., J.E. and S.M. (University of Bonn) and S.P.S. (University of Freiburg). This work was supported by the Deutsche Forschungsgemeinschaft (DFG, German Research Foundation) SFB 1381 project ID 403222702 (to U.S., N.W. and B.F.); PF 202/9-1 project ID 394024777 (to N.P.); FA 332/15-1 project ID 439189341, FA 332/17-1 project ID 446245862 and TRR 152 project ID 239283807 (to B.F.); BE 4679/2-2 project ID 269424439 and SFB 1218 project ID 269925409 (to T.B.); WI 4506/1-1 project ID 406757425 (to N.W.); Germany's Excellence Strategy (CIBSS EXC-2189 project ID 390939984, to U.S., H.R., N.W., N.P. and B.F.); and the European Research Council (ERC) Consolidator grant no. 648235 (to N.W.).

**Author contributions** U.S., H.R., N.W., N.P., B.F. and T.B. conceived and supervised the project. U.S., T.B., C.S.M., W.B., A.H. and B.F. acquired and evaluated data related to csBN–MS analyses and set-up MitCOM for the CEDAR platform. F.d.B., A.G., J.S., J.E., S.M., C.M., L.E., C.P., S.P.S., K.N.D., B.K., H.R., N.W., N.P. and T.B. performed and analysed experiments related to the analysis of protein assemblies in yeast. U.S., F.d.B., A.H., S.P.S., H.R., B.F. and T.B. prepared figures. B.F., U.S., T.B. and N.P. wrote the paper with the support of all of the authors.

**Competing interests** The authors declare no competing interests.

**Additional information**
**Correspondence and requests for materials** should be addressed to Nikolaus Pfanner or Bernd Fakler.

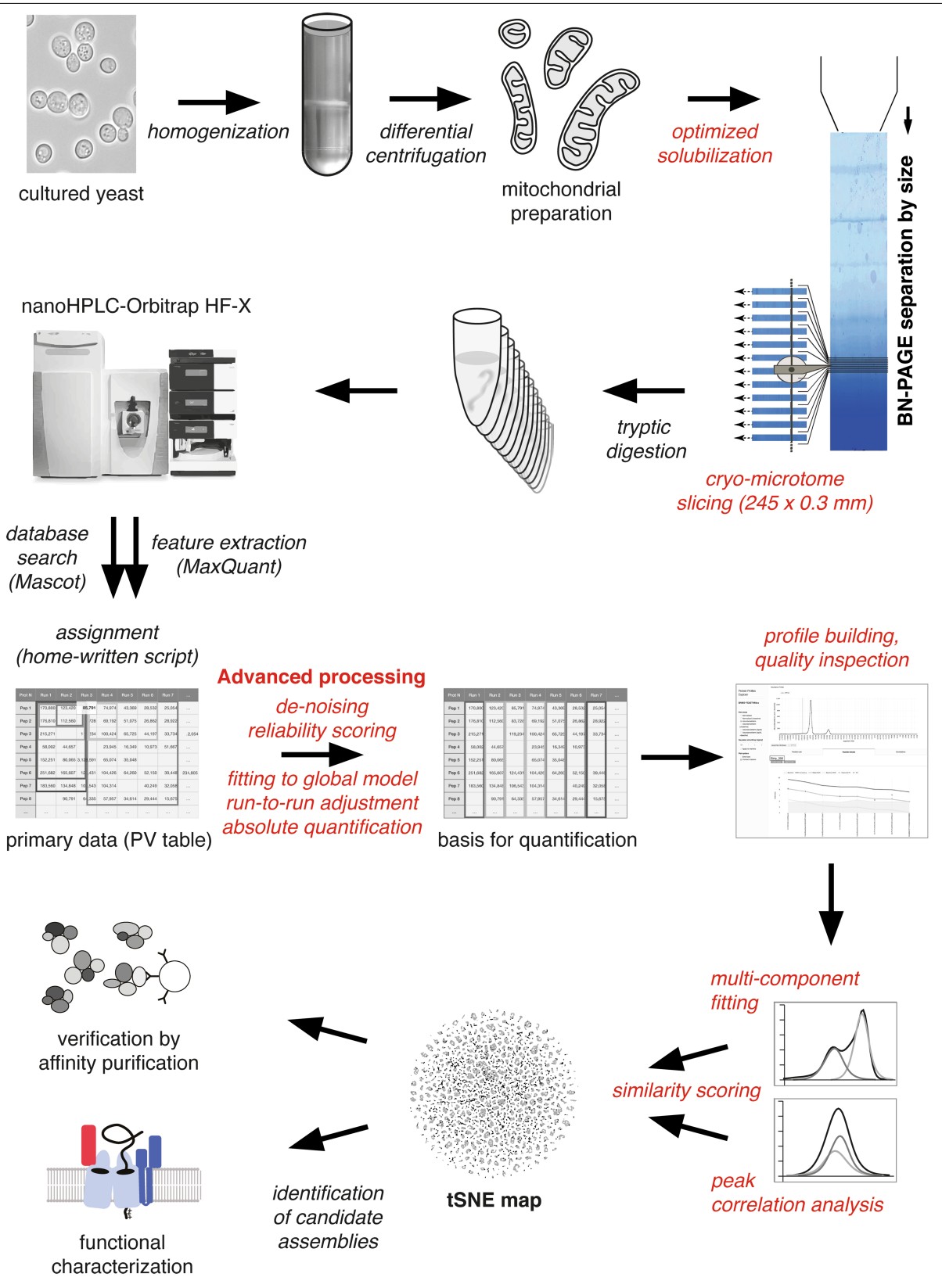

**Extended Data Fig. 1 | Experimental workflow for determination and evaluation of MitCOM.** Scheme illustrating major materials, tools and intermediates used to generate the MitCOM dataset and to identify protein complexes together with their subunit composition (for details see Methods). Novel or significantly improved tools or procedures are highlighted in red. Quality and usability of MitCOM benefitted particularly from the high-resolution sampling of the blue native gel, the advanced processing of MS-data for determination of accurate protein abundance–mass profiles and the new viewer toolbox including multi-Gaussian component fitting and correlation analysis. Note that the final assignment of protein profile peaks to a distinct complex requires biochemical verification by affinity purification and functional analysis. We used elements of Figs. 1a, 5h, Extended Data Figs. 3b, 5 and an adapted image of Thermo Scientific™ Q Exactive™ HF-X hybrid quadrupole-Orbitrap™ MS for illustration (permission by copyright owner: Thermo Fisher Scientific Inc., Waltham, MA, USA).

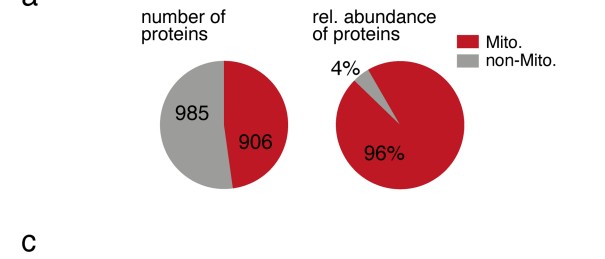

a

number of proteins

985 906

rel. abundance of proteins

4%

96%

Mito. non-Mito.

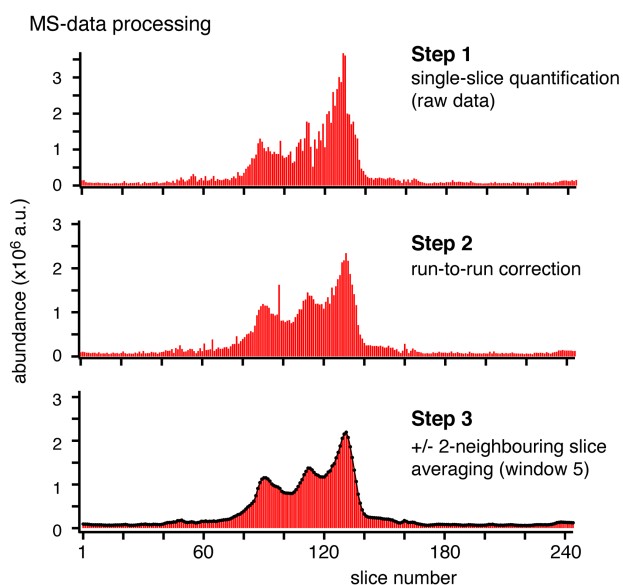

b MS-data processing

**Step 1**
single-slice quantification (raw data)

**Step 2**
run-to-run correction

**Step 3**
+/- 2-neighbouring slice averaging (window 5)

abundance (x10⁶ a.u.)

slice number

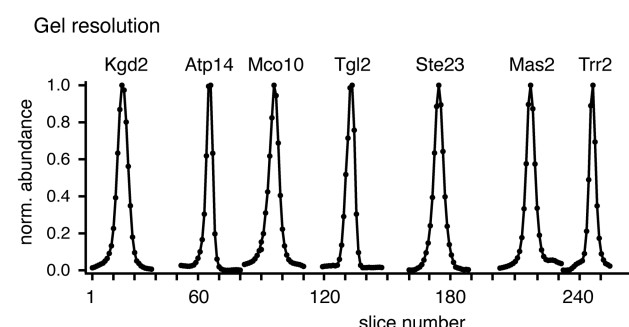

c Gel resolution

Kgd2  Atp14  Mco10  Tgl2  Ste23  Mas2  Trr2

norm. abundance

slice number

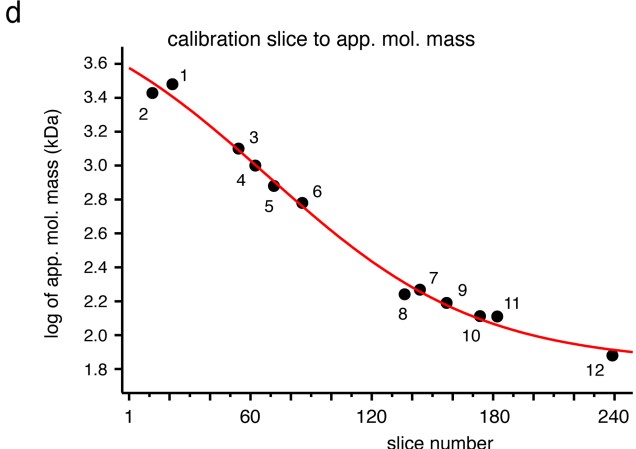

d calibration slice to app. mol. mass

log of app. mol. mass (kDa)

slice number

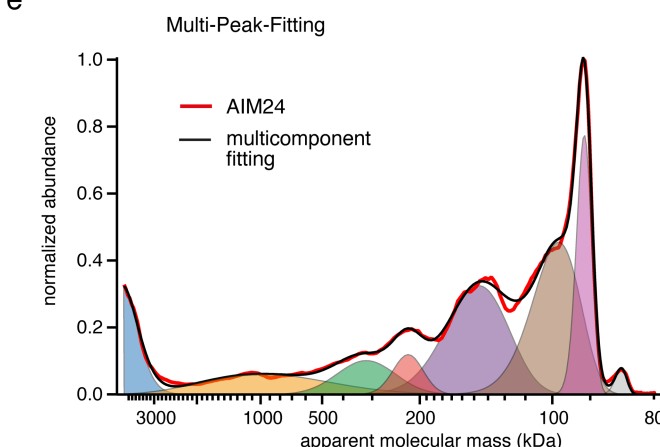

e Multi-Peak-Fitting

AIM24
multicomponent fitting

normalized abundance

apparent molecular mass (kDa)

**Extended Data Fig. 2 | Benchmarking of cryo-slicing blue native mass spectrometry. a**, Diagram summarizing number and abundance of proteins identified in the mitochondrial preparation from yeast. We used a mild procedure for mitochondrial preparation such that possible interaction partners from other cellular compartments remained associated with the mitochondrial surface/outer membrane complexes **b**, Stepwise processing of MS-data as detailed in Materials and Methods using protein Por2 (VDAC2) as an example; the data points and line in black at step 3 represent the 'abundance over slice number' profile for Por2. **c**, Profiles (around the abundance peak) of the indicated mitochondrial proteins illustrating equally high resolution over the entire blue native gel range. **d**, Conversion of slices to apparent molecular mass using fit of a sigmoidal function (curve in red) to the masses of the following proteins: 1, Oxoglutarate Dehydrogenase/ Ketoglutarate Dehydrogenase complex (3020 kDa); 2, pre-60S ribosome large

subunit (2680 kDa); 3, Dimer of $F_1F_O$-ATP synthase (1250 kDa); 4, $III_2IV_2$ supercomplex (1000 kDa); 5, $III_2IV_1$ supercomplex (750 kDa); 6, Monomer of $F_1F_O$-ATP synthase (600 kDa); 7, SAM-Mdm10 complex (185.5 kDa); 8, TIM22 complex (174 kDa); 9, ATM1 (155 kDa); 10, $SAM_{core}$ complex (Sam50, Sam37, Sam35, 129.4 kDa); 11, succinate dehydrogenase (130 kDa); 12, NADPH-cytochrome P450 reductase (Ncp1, 77 kDa). **e**, Multi-peak fitting by an automated algorithm approximating the profile data obtained for Aim24 (altered inheritance rate of mitochondria) (red line) by a sum of eight Gaussian functions (see Methods). The procedure provided the Gaussian parameters midpoint (apparent molecular mass), amplitude (abundance) and half-width (focusing or variation in size) for the peaks detected in a given profile. Black line represents the result of the multi-component fit, individual Gaussians are colour-coded.

## a    Raw data and 'Gaussian filtering'

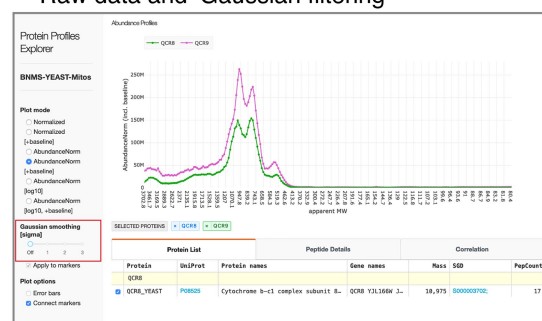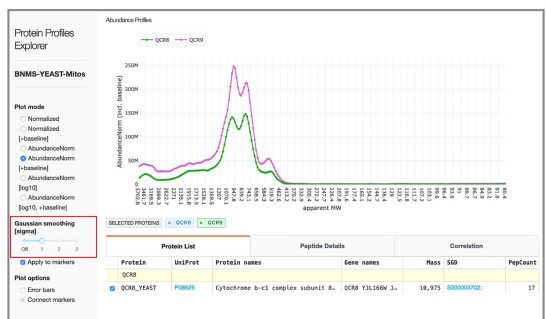

## b    Inspection of 'Peptide Details'

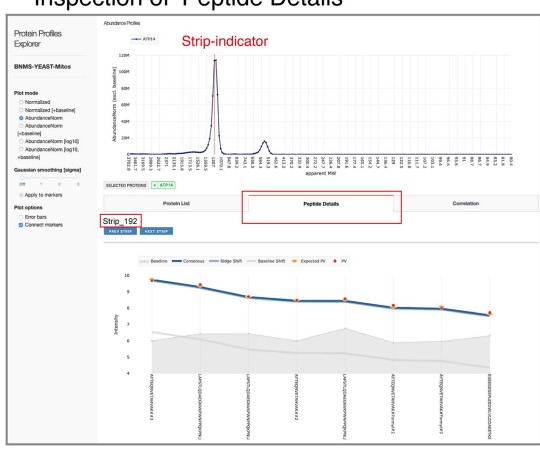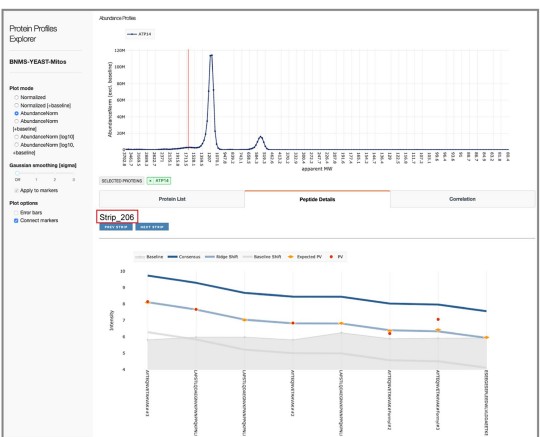

## c    Correlation analysis for selected complexes

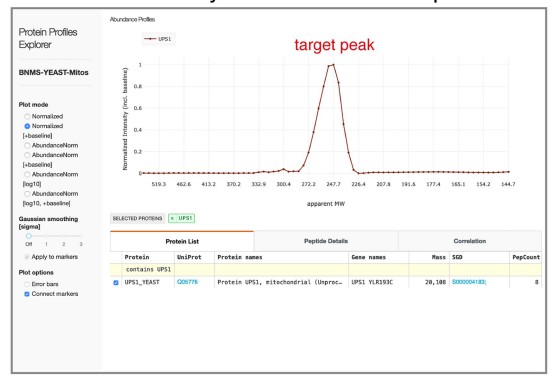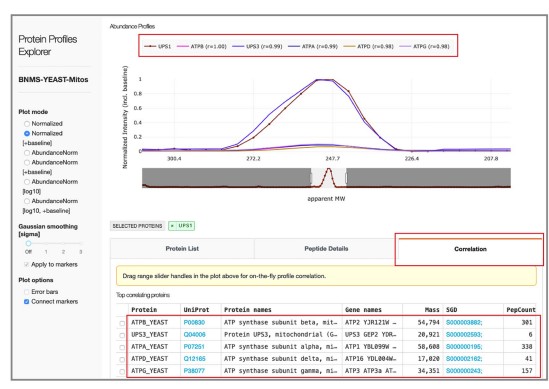

## d    Absolute and normalized abundances

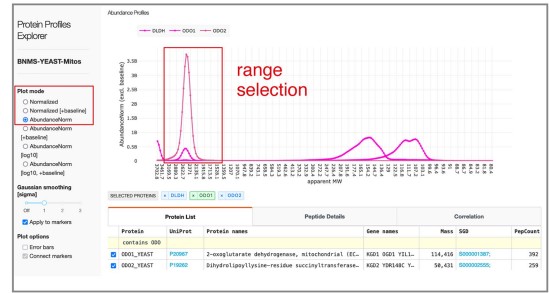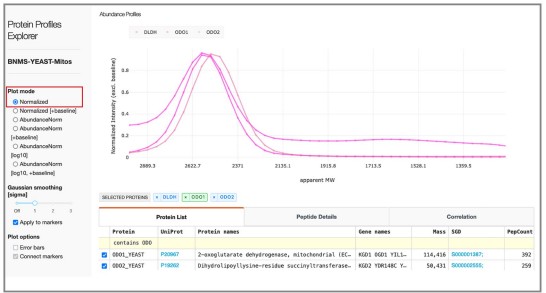

**Extended Data Fig. 3** | See next page for caption.

**Extended Data Fig. 3 | Using the Profile Viewer for interactive data exploration.** Illustration of a selected set of functions (highlighted in red) for targeted data inspection and analysis implemented in the MitCOM Profile Viewer (https://www3.cmbi.umcn.nl/cedar/browse/experiments/CRX36 or, with extended functions, at https://www.complexomics.org/datasets/mitcom). The features include profile visualization and inspection of quantification details, as well as custom-defined MitCOM-wide Pearson correlation analysis for searching constituents of protein complexes. **a**, adjustable Gaussian smoothing of displayed abundance–mass profiles; **b**, inspection of peptide details used for protein quantification (i.e. underlying the selected datapoint) and generation of the abundance–mass profile(s); **c**, correlation analysis tool. A region of interest is targeted ('target peak') within the profile of a selected protein (left). Correlation analysis is started by clicking the "correlation" tab that opens a range-selection panel in which left and right boundaries of the target peak can be set by dragging the range-slider handles. Simultaneously, a list of 20 proteins exhibiting the highest Pearson correlation with the selected target peak segment is displayed (right, calculation in real time); **d**, normalization of selected sets of proteins to their maximum. A help page with detailed user instructions is available on the Viewer's main window.

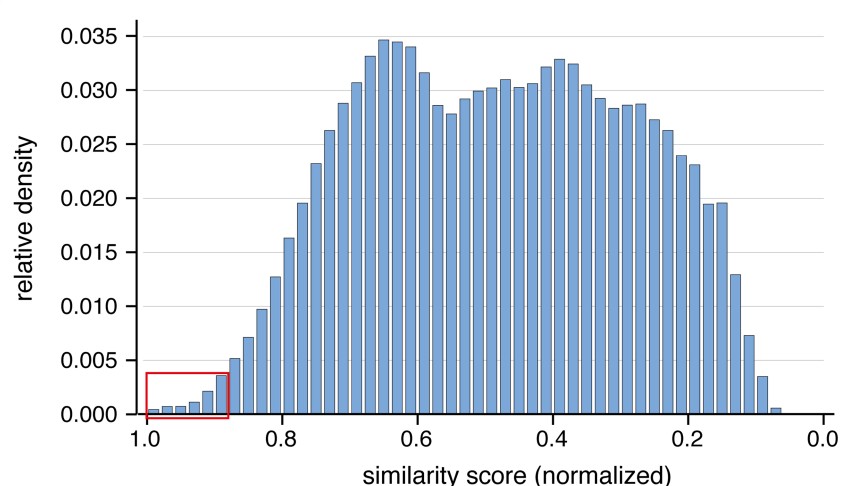

a

**b**

| Figure | Protein [Segment] Pair | | | SimScore |
|---|---|---|---|---|
| Fig. 1c | Phb1 [15-38] | – | Phb2 [15-38] | 0.998 |
| | Atp1 [76-88] | – | Atp2 [76-88] | 0.999 |
| | Atp1 [76-88] | – | Atp3 [76-88] | 0.999 |
| | Atp3 [76-88] | – | Atp2 [76-88] | 1.000 |
| | Ups1 [116-130] | – | Ups3 [116-130] | 0.993 |
| | Sdh1 [170-194] | – | Sdh2 [170-194] | 0.997 |
| Fig. 3a | Tom40 [101-122] | – | Om14 [101-122] | 0.973 |
| | Tom22 [126-140] | – | Om14 [126-140] | 0.976 |
| | Tom5 [126-140] | – | Om14 [126-140] | 0.979 |
| | Om14 [218-230] | – | Om45 [218-230] | 0.987 |
| Fig. 3c | Tom20 [128-140] | – | Pth2 [128-140] | 0.966 |
| | Tom40 [101-122] | – | Pth2 [100-122] | 0.955 |
| Fig. 4a | Tom5 [82-105] | – | Rsp5 [85-101] | 0.900 |
| | Tom5 [118-137] | – | Ubp16 [123-137] | 0.888 |
| ED Fig. 8a | Sam50 [132-156] | – | Sam35 [132-156] | 0.996 |
| | Sam50 [127-143] | – | Sam37 [127-143] | 0.996 |
| | Sam50 [127-143] | – | Mdm10 [127-143] | 0.956 |
| | Tom5 [106-117] | – | Sam50 [103-118] | 0.886 |
| | Tom5 [106-117] | – | Sam35 [102-117] | 0.884 |
| | Tom5 [106-117] | – | Mdm10 [110-116] | 0.885 |
| ED Fig. 8b | Tim12 [137-148] | – | Tim18 [137-148] | 0.984 |
| | Tim12 [137-148] | – | Tim22 [137-148] | 0.986 |
| | Tim12 [137-148] | – | Tim54 [137-148] | 0.977 |
| | Tim12 [120-150] | – | Tim9 [120-150] | 0.977 |
| | Tim12 [137-148] | – | Tim10 [137-148] | 0.962 |
| | Tim9 [201-221] | – | Tim10 [201-221] | 0.988 |
| | Sdh3 [170-194] | – | Sdh4 [170-194] | 0.998 |
| | Tim22 [137-148] | – | Sdh3 [137-148] | 0.980 |
| ED Fig. 8d | Atp1 [172-185] | – | Atp2 [172-185] | 0.998 |
| | Atp1 [76-88] | – | Atp2 [76-88] | 0.999 |
| | Atp1 [76-88] | – | Atp4 [76-88] | 0.996 |
| | Atp1 [77-93] | – | Atp5 [77-93] | 0.984 |
| | Atp7 [77-93] | – | Atp1 [77-93] | 0.998 |
| | Atp1 [77-93] | – | Atp1 [77-93] | 0.999 |
| | Atp21 [78-91] | – | Atp20 [78-91] | 0.994 |
| ED Fig. 8e | Phb1 [15-38] | – | Yta12 [14-39] | 0.980 |
| | Phb1 [15-38] | – | Yta10 [14-39] | 0.971 |
| | Phb1 [9-23] | – | Mdj2 [9-23] | 0.947 |
| | Phb2 [15-38] | – | Yta12 [14-39] | 0.980 |
| | Phb2 [15-38] | – | Yta10 [14-39] | 0.971 |
| | Phb2 [8-23] | – | Mdj2 [9-23] | 0.948 |
| ED Fig. 8g | Cox1 [46-75] | – | Cox2 [46-75] | 0.987 |
| | Cox1 [53-71] | – | Cox4 [53-71] | 0.986 |
| | Cox1 [53-71] | – | Cox6 [54-71] | 0.992 |
| | Cox1 [53-71] | – | Qcr2 [52-72] | 0.957 |
| | Cox1 [45-66] | – | Shy1 [45-64] | 0.921 |

**Extended Data Fig. 4 | Unsupervised matching analysis of MitCOM protein profile segments. a**, Histogram of similarity scores obtained for the segments of the abundance–mass profiles of MitCOM proteins (total of $1.2 \times 10^9$ values). The scores integrate (i) correlation of the profile segments of any MitCOM protein with those from all MitCOM proteins (local Pearson correlation, Extended Data Fig. 3c), (ii) ratio of their molecular abundance and (iii) distance of their maxima (see Methods); a value of 1.0 indicates perfect agreement of all three parameters. Pairs in groups of protein profile peaks shown in the figures and/or selected for biochemical verification experiments had score values between 1.0 – 0.88 (highlighted in red). **b**, Pairs of protein profile segments (indicated by the slice numbers given in brackets) exhibiting best similarity scores ('simscore') of the proteins highlighted and/or used for further experiments in the indicated figures.

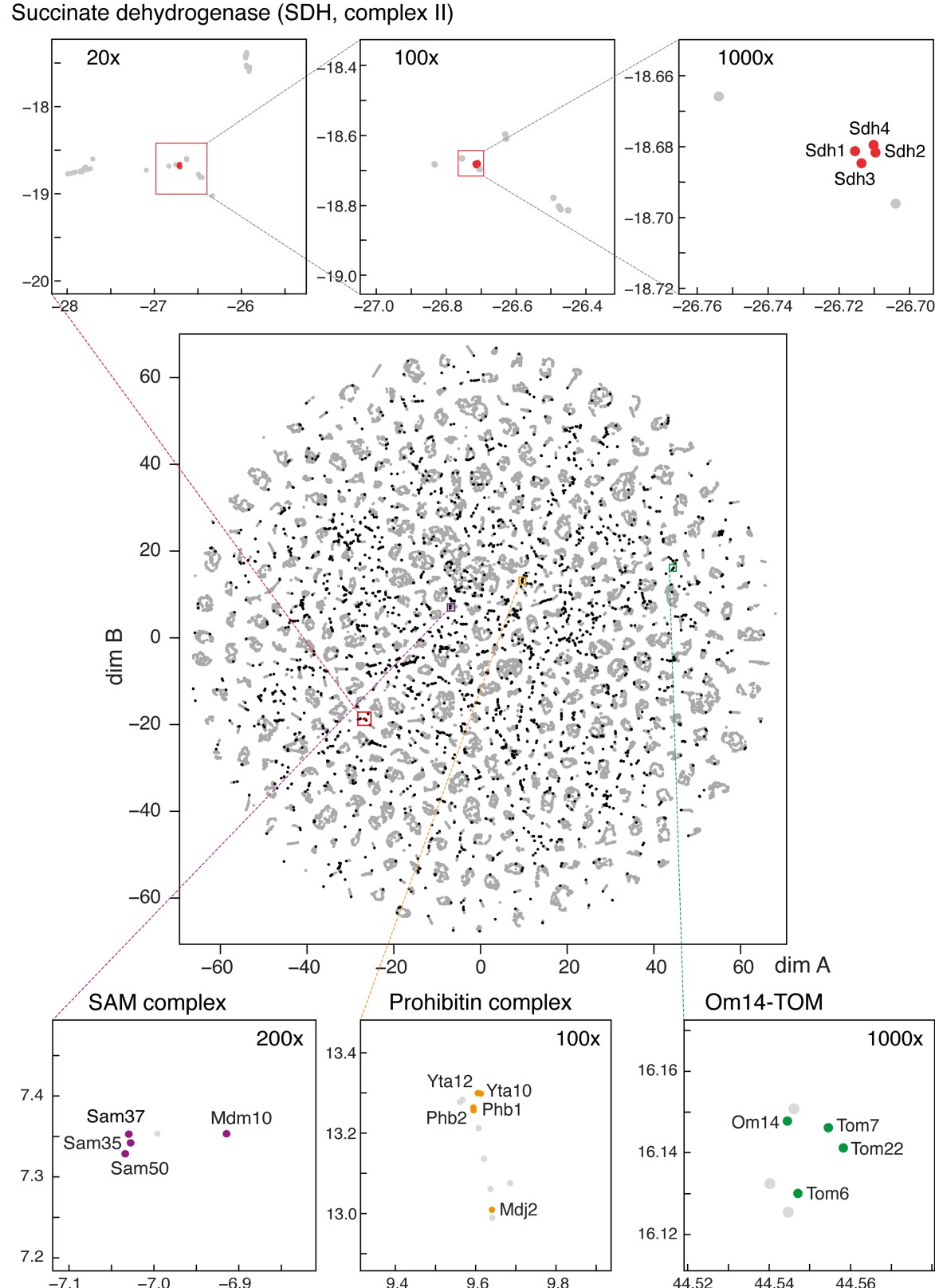

Succinate dehydrogenase (SDH, complex II)

20x

100x

1000x

Sdh4
Sdh1 Sdh2
Sdh3

dim B

dim A

SAM complex

200x

Sam37 Mdm10
Sam35
Sam50

Prohibitin complex

100x

Yta12 Yta10
Phb2 Phb1

Mdj2

Om14-TOM

1000x

Om14 Tom7
Tom22
Tom6

**Extended Data Fig. 5** | See next page for caption.

**Extended Data Fig. 5 | Visualization of the matching analysis within the MitCOM dataset.** Center, Plot showing t-distributed stochastic neighbourhood distribution (t-SNE) of the entirety of protein abundance profile segments (49,111) represented by dots. Results of the automated Gaussian-fitting (Fig. 1c) were used to define 'seeds' (black dots) for Pearson correlation with the corresponding sections of all 906 MitCOM protein profiles (see also Viewer tool in Extended Data Fig. 3c). Similarity scores (Methods and Extended Data Fig. 4) were calculated for all protein sections (components) with r > 0.95 (grey dots) and used as distance measure for t-SNE. As a result, close co-localization on this map indicates potential association of the respective proteins into one or more complexes with a defined apparent molecular mass (i.e. localization in Fig. 1c). Top insets are stepwise zooms into the plot resolving a closely co-localized group of protein profile segments representing the main peak of succinate dehydrogenase (respiratory chain complex II) assembled from four subunits Sdh1–4. Note the exquisite co-localization and discrimination of these subunit even at highest magnification factors (indicated in each inset). Bottom insets are magnification of the framed windows in the central plot demonstrating close co-localization of profile segments of biochemically verified subunits of the indicated multi-protein complexes: SAM complex (left), prohibitin/*m*-AAA supercomplex (middle), assembly of Om14 with TOM (right).

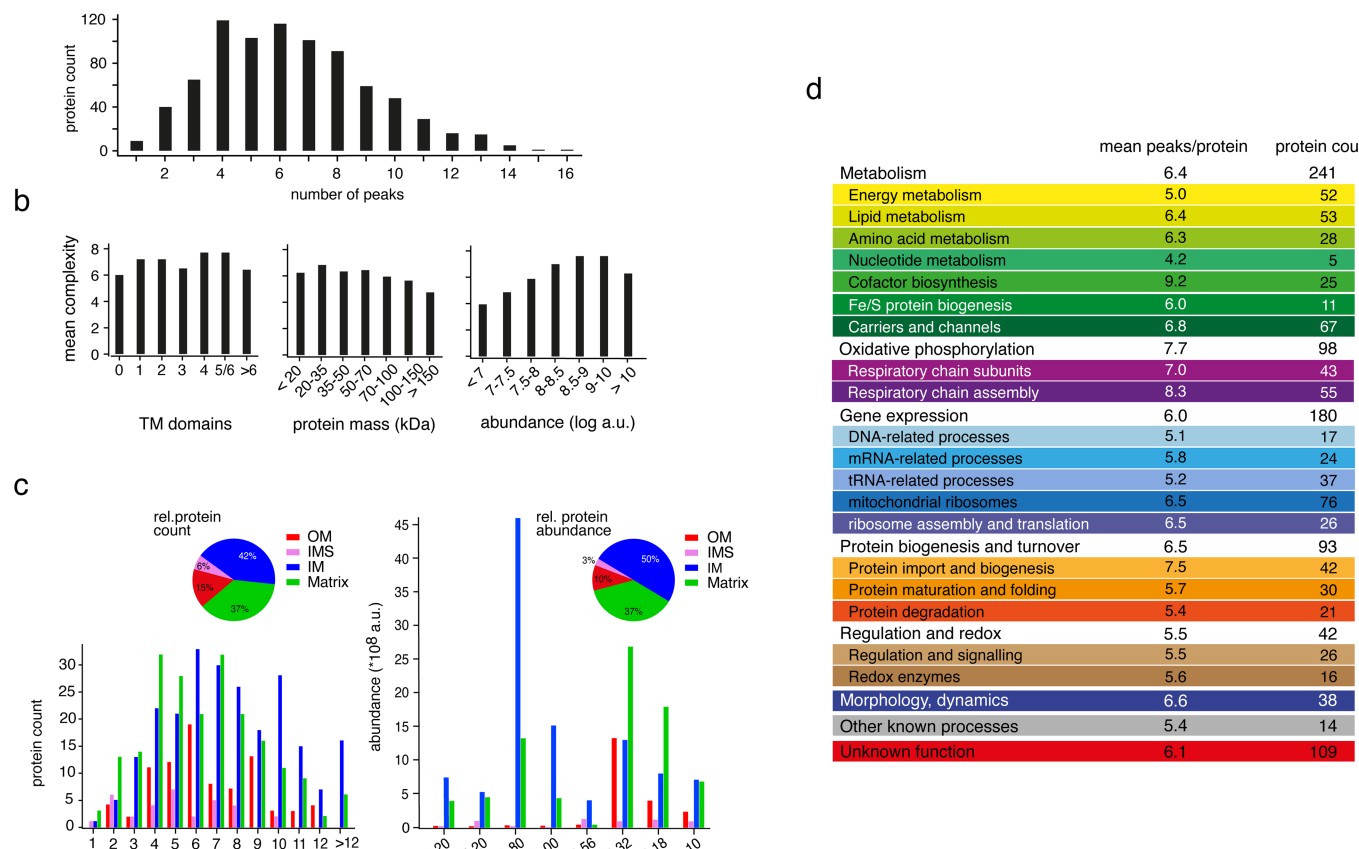

**Extended Data Fig. 6 | Complexity of mitochondrial protein assemblies.**
**a**, Complexity (number of peaks per protein) of mitochondrial assemblies identified in the abundance–mass profiles of the 818 mitochondrial proteins accessible to automated Gaussian fitting. The mean value of the complexity was 6.4 peaks/assemblies per protein (SEM: 0.1, n = 818 mitochondrial proteins). **b**, Mean complexity determined for proteins with the indicated number of predicted transmembrane domains (TM, left), protein mass (middle) and overall abundance in mitochondria (right). **c**, Complexity and size related to submitochondrial localization. Bar graphs summarizing the complexity (left panel, number of peaks per protein) or the apparent molecular size (right panel, mass range bins) of assemblies/complexes identified in the abundance–mass profiles of proteins located in the outer and inner membranes (OM, IM), the intermembrane space (IMS) or the matrix of mitochondria. The plots represent results of 562 proteins for which subcompartmental localization was extracted from SGD GO terms. Relative numbers (left) and abundances (right) of the proteins in the subcompartments are indicated by pie charts. The apparent mass range was binned into equal log-value intervals (bins of 0.25) of the apparent molecular mass. The highest number of peaks (complexity) was observed for inner membrane proteins. The molecular mass ranges of protein assemblies strongly differ between the mitochondrial subcompartments. Inner membrane complexes are present in high abundance in the mass range of ~500–1,800 kDa, consistent with the presence of oxidative phosphorylation supercomplexes and further large membrane protein complexes[8,74], whereas outer membrane complexes, including protein translocases and metabolite channels, and many matrix complexes display the highest abundance between 100–320 kDa. **d**, Complexity related to functional categories. Mean complexity of different functional categories and subcategories of mitochondrial proteins. Protein count, number of proteins representing these categories in the complexome determined in this work. Regarding the molecular mass distribution of protein assemblies, the two most abundant functional categories, oxidative phosphorylation and metabolism, display major differences (Fig. 2). Proteins involved in oxidative phosphorylation show the highest abundance in a molecular mass range of ~500–1,800 kDa. A fraction of proteins operating in metabolic processes migrate in the high molecular mass range >1,800 kDa, the majority of metabolism-linked proteins, however, migrate below 320 kDa. The category protein biogenesis and turnover shows the highest abundance between 180 and 320 kDa in agreement with the sizes observed for major protein translocases[62,70,71]. Proteins involved in regulatory or redox processes are predominantly found in the low molecular mass range. The category morphology and dynamics displays a broad distribution with a maximum at an apparent mass of >3,200 kDa (Fig. 2) that includes the mitochondrial contact site and cristae organizing system (MICOS) and associated assemblies[1,10].

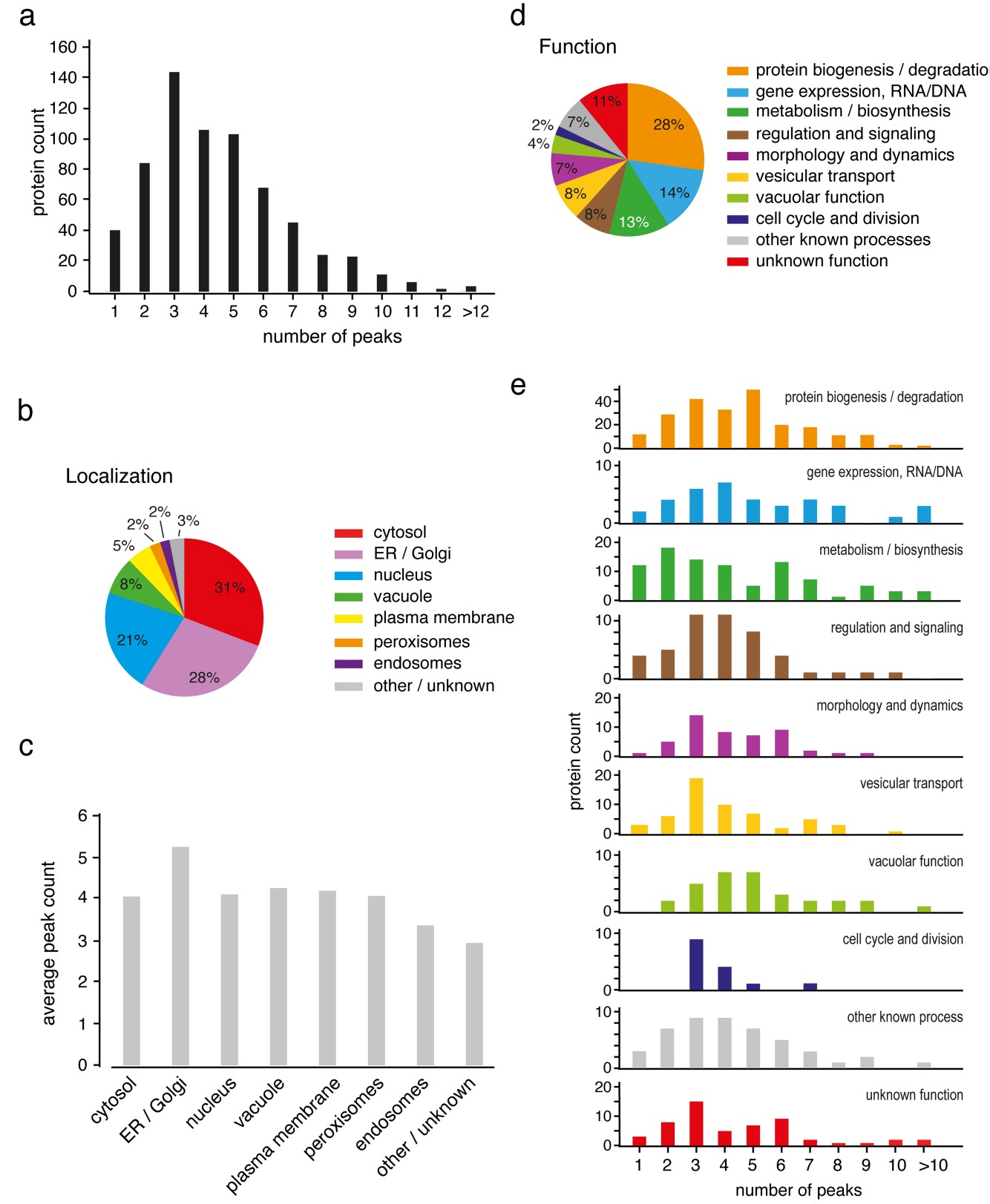

**Extended Data Fig. 7 | Localization, complexity and function of non-mitochondrial proteins. a**, Bar graph summarizing the complexity (manually verified number of peaks per protein) for the 660 non-mitochondrial proteins accessible to profile evaluation as in Fig. 1c. **b**, Distribution of subcellular localizations annotated for the 985 non-mitochondrial proteins identified by MS-analysis in the mitochondrial preparation used. Note that the majority of these proteins localized either to cytosol or to ER / Golgi. **c**, Mean complexity determined for the proteins in the indicated subcellular compartments. **d**, Functional categorization of non-mitochondrial proteins according to their annotation in Uni-Prot/Swissprot and SGD. Functional categories from Fig. 2 were used when applicable. **e**, Bar graph summarizing the complexity determined for the proteins of the indicated functional categories.

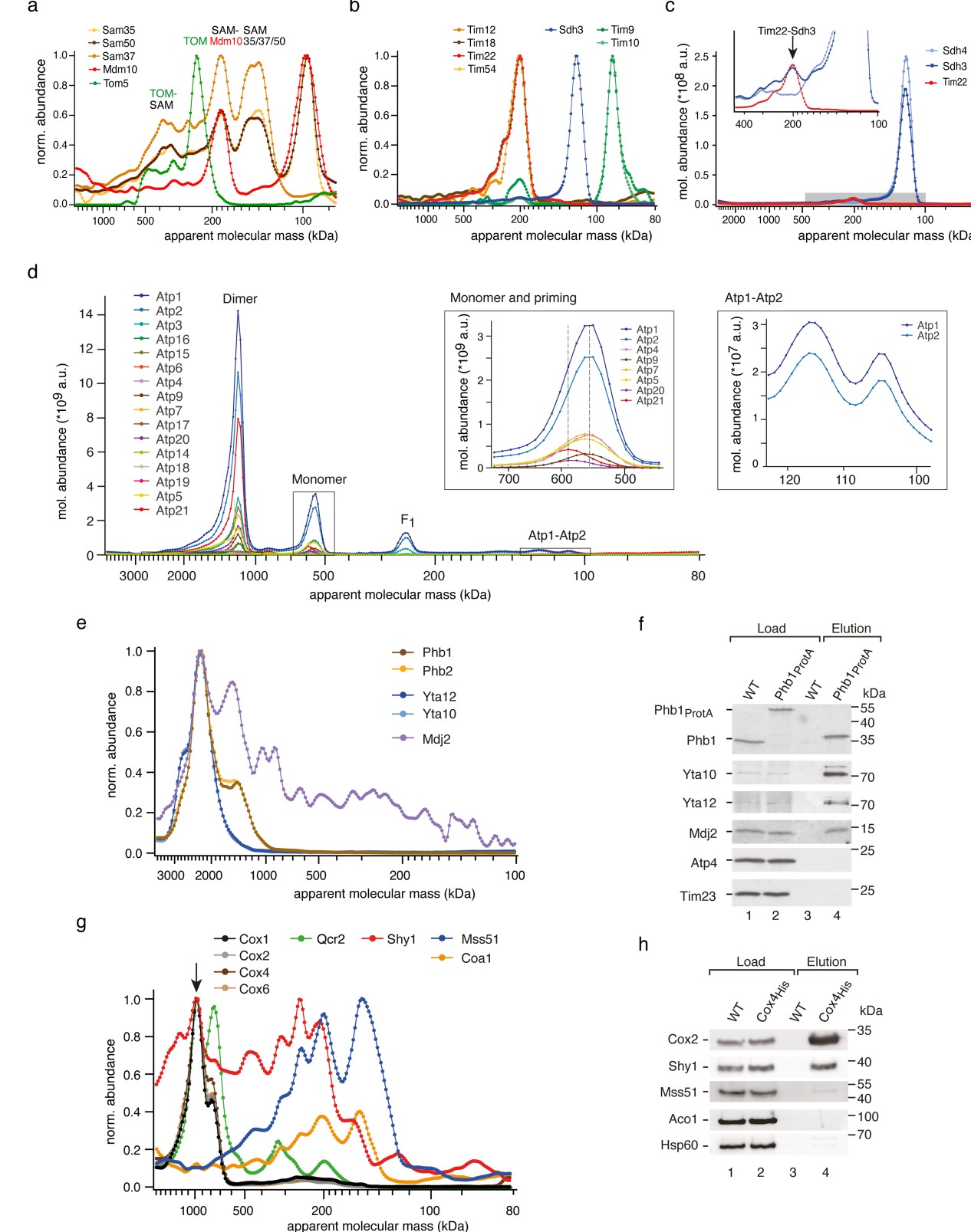

**Extended Data Fig. 8 |** See next page for caption.

**Extended Data Fig. 8 | Characterization of dynamic protein machineries by MitCOM. a**, Analysis of the SAM complex illustrates the application of MitCOM in the characterization of dynamic protein machinery. The figure shows normalized abundance–mass profiles of SAM subunits, Mdm10 and Tom5. Distinct forms of SAM and TOM complexes are indicated. The SAM complex, also termed TOB complex for topogenesis of β-barrel proteins, inserts the precursors of β-barrel proteins into the mitochondrial outer membrane. SAM consists of several forms that are in exchange with each other and couple to different partner proteins[62,75-79]. The $SAM_{core}$ complex consists of Sam50, Sam37 and Sam35[62,65,77]. Coupling to TOM (transient TOM-SAM supercomplex) facilitates the transfer of precursor proteins from TOM to SAM, whereas the coupling to Mdm10 (SAM-Mdm10 complex) promotes precursor release[62,76,78,80]. In addition, Sam37 plays a dynamic role[62] and its release from $SAM_{core}$ leads to a Sam35-Sam50 subcomplex[81,82]. The apparent overlap of the Sam35-Sam50 subcomplex with Mdm10 in the low molecular mass range does not reflect co-assembly as the peaks can be distinguished by the differences in peak maxima and half-widths. MitCOM efficiently resolves the distinct SAM forms: Sam 35-Sam50, $SAM_{core}$, SAM-Mdm10, and TOM-SAM supercomplexes.
**b**, **c**, The interconnected system of the TIM22 complex, the small TIM chaperones Tim9-Tim10, and the SDH complex illustrates a high precision of MitCOM in separation and analysis of complexes of different abundance. The figures show abundance–mass profiles of carrier pathway TIM proteins and SDH (inset in (c): grey box at enlarged scale). The TIM22 complex[63,83] migrates in a main peak of ~200 kDa in MitCOM. For Tim9 and Tim10, two populations are resolved, one bound to the TIM22 complex and the other one representing the soluble hetero-hexameric chaperone complex of the intermembrane space. Sdh3 is found in the SDH complex and the ~20-times less abundant TIM22 complex (c-normalized abundance vs. b-absolute abundance)[6,63,84]. Note that the precursor of Sdh3 is imported and inserted into the inner membrane via the presequence translocase TIM23 and the OXA translocase, not via the TIM22 complex; mature Sdh3 subsequently assembles into the SDH complex and the TIM22 complex[85]. Sdh4 that is not part of the TIM22 complex does not display a peak at the TIM22 complex (c, insert). MitCOM thus efficiently and quantitatively separates TIM22, Tim9-Tim10 and SDH. **d**, MitCOM quantitatively separates distinct complexes of the $F_1F_0$-ATP synthase. The figure shows full-range abundance profiles of the subunits of $F_1F_0$-ATP synthase. The major peak at 1,200 kDa represents the fully assembled dimeric ATP synthase predominating under the mild solubilization conditions used (see Methods), followed by the monomer (at 560 kDa) and the $F_1$-module (at 250 kDa). Left inset,

discrimination of two monomeric ATP synthase complex populations. The abundant peak at 560 kDa lacks the dimerizing subunits Atp20 and Atp21. The primed monomeric form at 590 kDa includes Atp20 and Atp21[49]; note the shoulder-like appearance of the profile peaks of the common subunits caused by the overlap of these two assemblies. Right inset, close co-migration of Atp1 and Atp2 in the low molecular mass range. **e**, Normalized abundance–mass profiles of prohibitins Phb1 and Phb2, of the *m*-AAA protease subunits Yta10 and Yta12, and of the inner membrane-bound J-protein Mdj2. The prohibitin 1/2 (Phb1/2) complex forms an oligomeric scaffold and associates with the protease complex *m*-AAA (Yta10/12)[74]. In mammalian mitochondria, the J-protein DNAJC19 interacts with the prohibitin complex[74], yet it has been unknown if and which yeast mitochondrial J-protein may associate with the prohibitin complex. The inner membrane-bound J-protein Mdj2 migrates in multiple peaks, the largest and most abundant one perfectly co-migrating with the large prohibitin/*m*-AAA supercomplex of ~2.2 MDa. **f**, Affinity purification with tagged Phb1 demonstrates that Mdj2 is the yeast J-protein partner of the prohibitin/*m*-AAA supercomplex. Wild-type (WT) and $Phb1_{ProtA}$ mitochondria were subjected to affinity purification using Protein A sepharose. Bound proteins were eluted with TEV protease and analysed by immunodetection with the indicated antisera. Load 6%, elution 100%. Control proteins: Atp4, subunit of the $F_1F_0$-ATP synthase; Tim23, subunit of the presequence translocase of the inner membrane. **g**, Normalized abundance–mass profiles of structural subunits of respiratory complex IV (Cox1, Cox2, Cox4, Cox6) and complex III (cytochrome $bc_1$ complex; subunit Qcr2), and of the assembly factors Shy1, Coa1 and Mss51. Each of the assembly factors Coa1, Mss51 and Shy1 migrates in multiple peaks in line with their reported presence in various assembly intermediates[86-89]. A major peak of Shy1 tightly co-migrates with the fully assembled $III_2IV_2$ respiratory supercomplex of 1 MDa (arrow), in contrast to Coa1 and Mss51. **h**, WT and $Cox4_{His}$ mitochondria were subjected to affinity purification via Ni-NTA agarose and analysed by immunodetection with the indicated antisera. Load 2%, elution 100%. Control proteins: Aco1, aconitase; Hsp60, heat shock protein of 60 kDa. Tagged Cox4 that is not part of early Mss51-containing assembly intermediates[42] co-purifies Shy1 together with the structural subunit Cox2. Whereas qualitative blue native studies suggested that only a small fraction of Shy1 is associated with the full $III_2IV_2$ supercomplex[88-90], MitCOM provides a direct quantitative analysis and demonstrates that a main peak of Shy1 is present at the fully assembled supercomplex, different from other assembly factors such as Mss51 and Coa1.

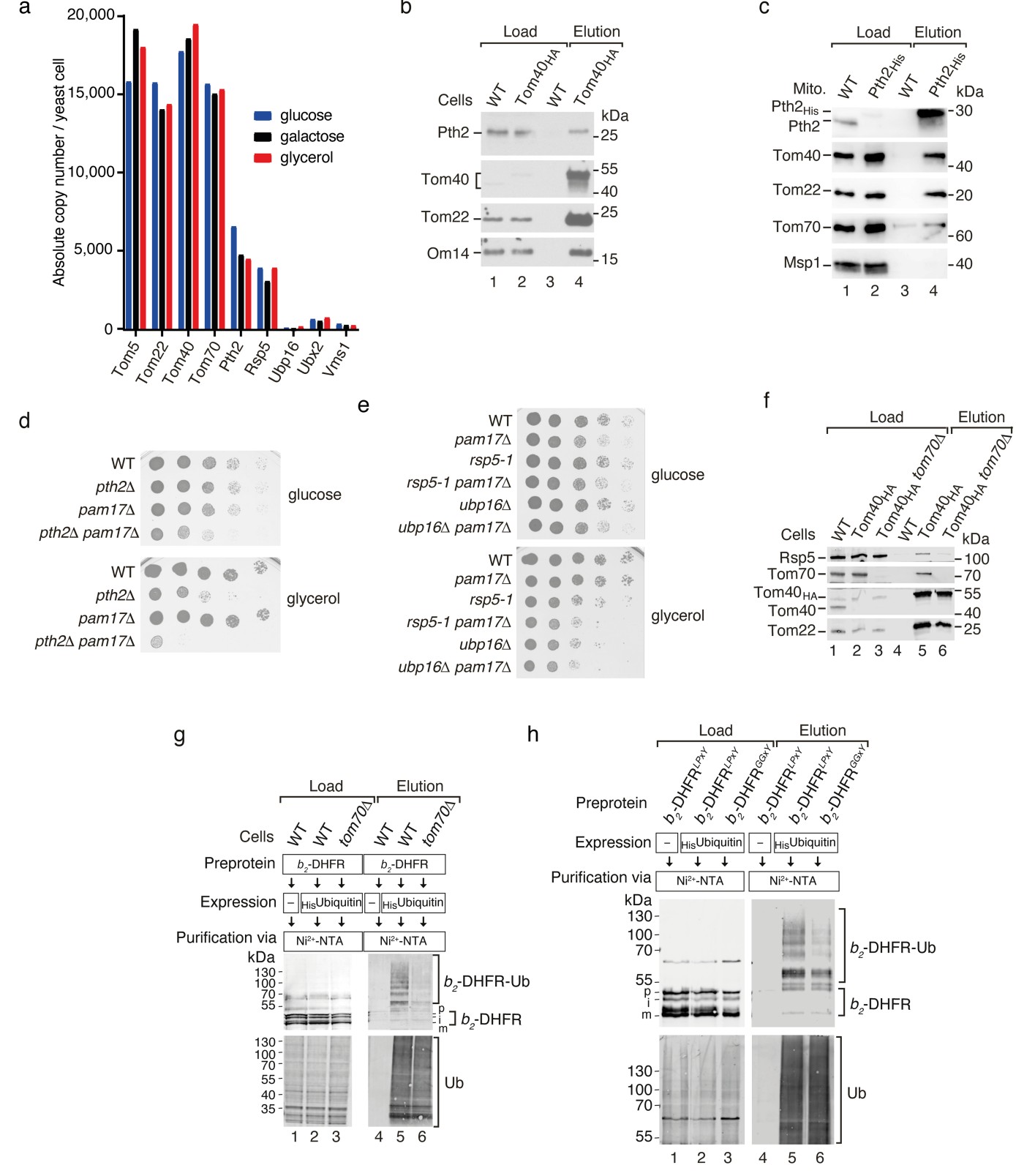

**Extended Data Fig. 9** | See next page for caption.

**Extended Data Fig. 9 | Characterization of Pth2, Ubp16 and Rsp5. a**, Absolute copy number per yeast cell of the indicated proteins. Values are based on the quantification of mitochondrial proteins grown on different carbon sources by Morgenstern et al.[6]. **b**, Wild-type (WT) and Tom40$_{HA}$ cell extracts were lysed with digitonin and subjected to affinity purification. Proteins were analysed by SDS-PAGE and immunodetection with the indicated antisera. Load 1%, elution 100%. **c**, WT and Pth2$_{His}$ mitochondria were lysed with digitonin and subjected to affinity purification via Ni-NTA agarose. Proteins were analysed by SDS-PAGE and immunodetection with the indicated antisera. Load 0.2% (Tom40, Tom22, Tom70) or 2% (Pth2, Msp1), elution 100%. Control protein: Msp1, mitochondrial sorting of proteins 1. The precursor of Pth2 is imported from the cytosol via the receptor Tom70 and inserted into the outer membrane by the mitochondrial import complex MIM[21]. **d**, Serial dilutions of the indicated strains were spotted onto full medium with either glucose or glycerol as carbon source and grown at 30 °C. **e**, Serial dilutions of the indicated strains were spotted onto full medium containing glucose or glycerol as carbon source and grown at 24 °C. **f**, Rsp5 preferentially binds to substrates and adaptor proteins via interaction with a PPxY/LPxY motif[31,91–93]. Tom70 contains a PPxY motif (aa 71–74) and may thus be involved in recruiting Rsp5 to mitochondria. WT, Tom40$_{HA}$ and Tom40$_{HA}$ *tom70Δ* cell extracts were lysed with digitonin and subjected to affinity

purification. Proteins were analysed by SDS-PAGE and immunodetection with the indicated antisera. Load 0.2%, elution 100%. The interaction of Rsp5 with the TOM complex is impaired in the absence of Tom70. **g**, WT and *tom70Δ* cells expressing $b_2$-DHFR and His-tagged ubiquitin as indicated were lysed under denaturing conditions and subjected to affinity purification via Ni-NTA agarose. Proteins were analysed by SDS-PAGE and immunodetection with antisera against the DHFR domain (upper panel) or ubiquitin (lower panel). $b_2$-DHFR-Ub, ubiquitin modified $b_2$-DHFR. Load 0.2%, elution 100%. Precursor ubiquitylation is diminished in the absence of Tom70. **h**, We found an LPxY motif in the DHFR domain of $b_2$-DHFR. WT yeast cells expressing the standard precursor cytochrome $b_2$-DHFR$^{LPxY}$ or the mutant form $b_2$-DHFR$^{GGxY}$ and His-tagged ubiquitin as indicated were lysed under denaturing conditions and subjected to affinity purification via Ni-NTA. Proteins were analysed by SDS-PAGE and immunodetection with the indicated antisera. $b_2$-DHFR-Ub, ubiquitin modified $b_2$-DHFR. Input 0.2%, elution 100%. Ubiquitylation of the mutant precursor was reduced. These data suggest that Tom70 is involved in recruiting Rsp5 to the mitochondrial import site. Rsp5 may use Tom70 as an adapter for interacting with precursor proteins and, in case a precursor substrate contains a PPxY/LPxY motif, also engage the motif on the precursor.

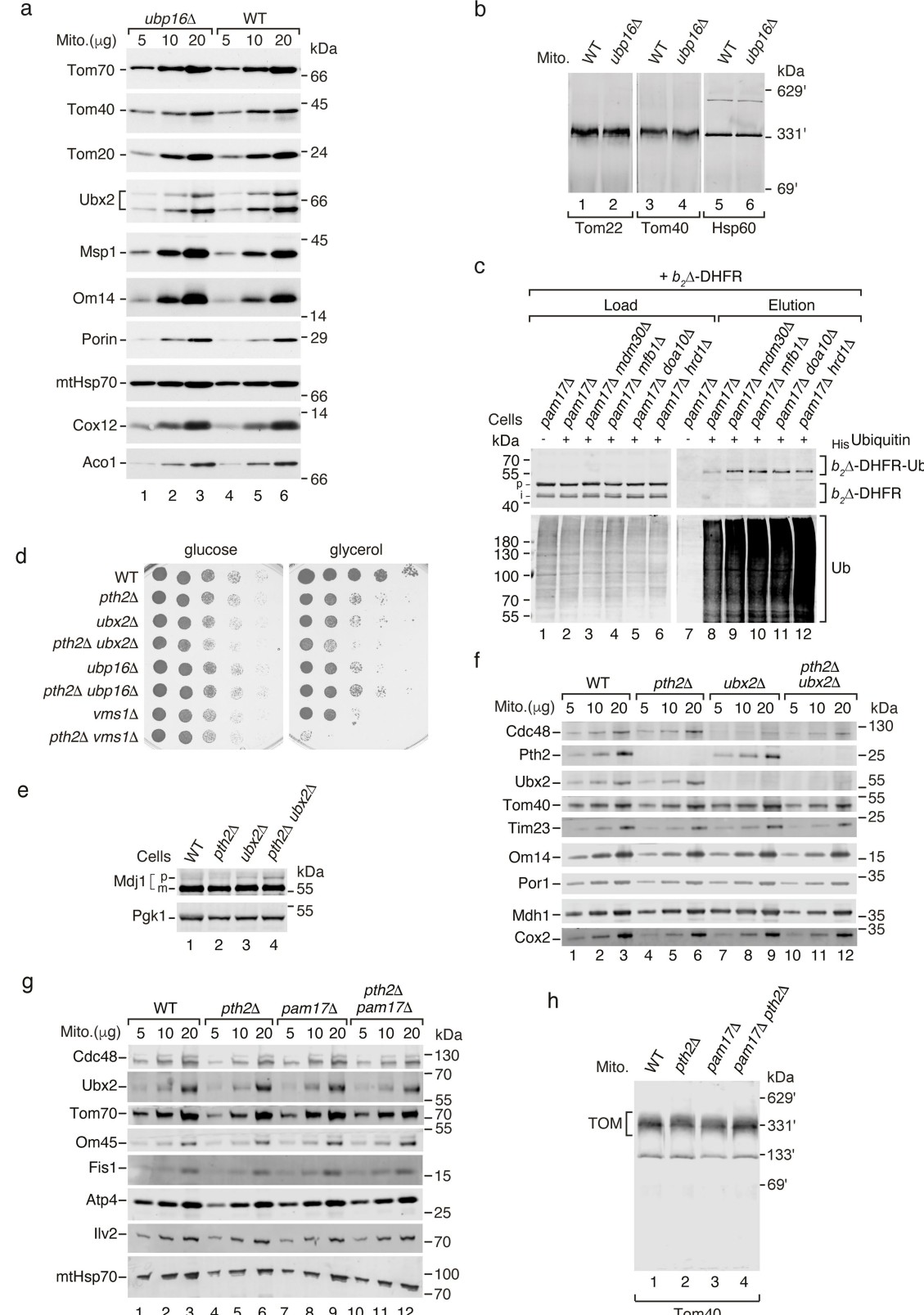

**Extended Data Fig. 10 |** See next page for caption.

**Extended Data Fig. 10 | Characterization of yeast cells and mitochondria lacking Ubp16, Pth2 or Ubx2. a**, The indicated protein amounts of WT and *ubp16Δ* mitochondria were analysed by SDS-PAGE and immunodetection with the indicated antisera. **b**, WT and *ubp16Δ* mitochondria were lysed with digitonin and analysed by blue native electrophoresis and immunodetection using an antiserum against Tom22, Tom40 and Hsp60. The levels of TOM, Ubx2 and further tested proteins (a) were not altered in *ubp16Δ* mitochondria, excluding that the lack of Ubp16 led to indirect effects on the protein levels of import or mitoTAD components. **c**, WT and the indicated strains expressing cytochrome $b_2$Δ-DHFR and His-tagged ubiquitin as indicated were lysed under denaturing conditions and subjected to affinity purification via Ni-NTA. Proteins were analysed by SDS-PAGE and immunodetection with the indicated antisera. $b_2$Δ-DHFR-Ub, ubiquitin modified $b_2$Δ-DHFR. Input 0.2%, elution 100%. **d**, Serial dilutions of the indicated strains were spotted onto full medium with either glucose or glycerol as carbon source and grown at 30 °C. **e**, Cell extracts of wild-type (WT), *pth2Δ*, *ubx2Δ* and *pth2Δ ubx2Δ* strains were analysed by immunodetection with the indicated antisera. p, precursor; m, mature form of Mdj1. The single deletions of Pth2 and Ubx2 did not lead to an accumulation of the Mdj1 precursor when the complete mitochondrial import machinery including Pam17 was present (see also Fig. 3d)[17]. However, the Mdj1 precursor moderately accumulated in *pth2Δ ubx2Δ* double mutants, supporting the view that a fraction of preproteins accumulate even when the mitochondrial import system is fully active[94–98]. Thus, at least one of the pathways, Ubx2 or Pth2, has to be active for removal of accumulated precursor proteins. **f**, The indicated protein amounts of WT, *pth2Δ*, *ubx2Δ* and *pth2Δ ubx2Δ* mitochondria were analysed by SDS-PAGE and immunodetection with the indicated antisera. **g**, The indicated protein amounts of WT, *pth2Δ*, *pam17Δ* and *pth2Δ pam17Δ* mitochondria were analysed by SDS-PAGE and immunodetection with the indicated antisera. **h**, WT, *pth2Δ*, *pam17Δ* and *pth2Δ pam17Δ* mitochondria were lysed with digitonin and analysed by blue native electrophoresis and immunodetection with Tom40 specific antisera. The steady state levels of Ubx2, Cdc48 and TOM are not substantially altered in *pth2Δ* and *pth2Δ pam17Δ* mutant mitochondria, excluding that the accumulation and ubiquitylation of precursor proteins (Fig. 3d, e) is indirectly caused by a loss of mitoTAD or TOM components. Yeast strains, plasmids and antisera are listed in Supplementary Tables 4–6 (refs. [99–102]).

Bernd Fakler

# Reporting Summary

## Statistics

For all statistical analyses, confirm that the following items are present in the figure legend, table legend, main text, or Methods section.

| n/a | Confirmed | |
|---|---|---|
| ☐ | ☒ | The exact sample size (*n*) for each experimental group/condition, given as a discrete number and unit of measurement |
| ☒ | ☐ | A statement on whether measurements were taken from distinct samples or whether the same sample was measured repeatedly |
| ☒ | ☐ | The statistical test(s) used AND whether they are one- or two-sided *Only common tests should be described solely by name; describe more complex techniques in the Methods section.* |
| ☒ | ☐ | A description of all covariates tested |
| ☒ | ☐ | A description of any assumptions or corrections, such as tests of normality and adjustment for multiple comparisons |
| ☐ | ☒ | A full description of the statistical parameters including central tendency (e.g. means) or other basic estimates (e.g. regression coefficient) AND variation (e.g. standard deviation) or associated estimates of uncertainty (e.g. confidence intervals) |
| ☒ | ☐ | For null hypothesis testing, the test statistic (e.g. *F*, *t*, *r*) with confidence intervals, effect sizes, degrees of freedom and *P* value noted *Give P values as exact values whenever suitable.* |
| ☒ | ☐ | For Bayesian analysis, information on the choice of priors and Markov chain Monte Carlo settings |
| ☒ | ☐ | For hierarchical and complex designs, identification of the appropriate level for tests and full reporting of outcomes |
| ☐ | ☒ | Estimates of effect sizes (e.g. Cohen's *d*, Pearson's *r*), indicating how they were calculated |

*Our web collection on statistics for biologists contains articles on many of the points above.*

## Software and code

Policy information about availability of computer code

| Data collection | LAS-3000, Amersham Imager 680 and Odyssey CLx for Western blot signals<br>QExactive HF-X mass spectrometer for acquisition of MS-data |
|---|---|
| Data analysis | Image Studio 5.2.5, ImageJ 2.1.0 and Adobe Photoshop 2021 for processing of data<br>Adobe Illustrator 2021 for figure preparation<br>msconvert v3.0.11098 for processing of primary MS data<br>MaxQuant 1.6.17 for calibration and quantification of MS data<br>Mascot 2.7 for database search (SwissProt_YEAST_20201007 + cRAP_20190304)<br>Igor Pro 9 (Wavemetrics) for data fitting and Figure preparation<br>Complexomics-mitcom 1.0 (Python package for peak detection, similarity score calculation, t-SNE visualization) available at https://doi.org/10.5281/zenodo.7355040 |

For manuscripts utilizing custom algorithms or software that are central to the research but not yet described in published literature, software must be made available to editors and reviewers. We strongly encourage code deposition in a community repository (e.g. GitHub). See the Nature Portfolio guidelines for submitting code & software for further information.

## Data

Policy information about availability of data

All manuscripts must include a data availability statement. This statement should provide the following information, where applicable:
- Accession codes, unique identifiers, or web links for publicly available datasets
- A description of any restrictions on data availability
- For clinical datasets or third party data, please ensure that the statement adheres to our policy

The mass spectrometric (MS) data generated during this study are available via ProteomeXchange with identifier PXD029548 https://doi.org/10.6019/PXD029548. UniProtKB/Swiss-Prot (https://www.uniprot.org; SwissProt_YEAST_20201007) and Saccharomyces Genome Database (SGD, https://www.yeastgenome.org; GO term mapping 20201027) were used as resource for structural, cell biological and functional annotation of proteins. The MitCOM datasets including an interactive profile viewer are available via https://www.complexomics.org/datasets/mitcom or the CEDAR platform https://www3.cmbi.umcn.nl/cedar/browse/experiments/CRX36. All other data are available in the main figures, Extended Data and Supplementary Information, including uncropped versions of gels/blots in Supplementary Fig. 1, and Supplementary Tables 1-3 (Excel files).

# Field-specific reporting

Please select the one below that is the best fit for your research. If you are not sure, read the appropriate sections before making your selection.

☒ Life sciences ☐ Behavioural & social sciences ☐ Ecological, evolutionary & environmental sciences

For a reference copy of the document with all sections, see nature.com/documents/nr-reporting-summary-flat.pdf

# Life sciences study design

All studies must disclose on these points even when the disclosure is negative.

| | |
|---|---|
| Sample size | The sample size was determined based on previous experience with specific types of experiments like the amount of mitochondria used for steady state and pulldown analysis (Martensson et al., 2019, Priesnitz et al., 2021). According to this the required amount of mitochondrial or cellular proteins was selected for each experiment. For key experiments, several runs were performed to determine the optimal sample size. The sample size or reference of sample size of each experiment is stated in the the Methods section. |
| Data exclusions | No data were excluded from this study. All relevant data are shown. |
| Replication | Representative images are shown for growth and biochemical assays/western blotting, including analysis of yeast growth (wild-type and mutants), total cell extracts, affinity purification from cell extracts, subcellular fractionation, protein steady state levels, blue native electrophoresis and affinity purification from isolated mitochondria. The findings were confirmed by independent experiments; in the main figures this applies to the following figures (minimum number of independent experiments in parentheses): 3b (2), 3d (3), 3e (2), 4b (2), 4c (2), 4d (2), 4e (2), 4f (2), 4g (2), 5a (2), 5b (3), 5c (2), 5d (3), 5e (2), 5f (3), and 5g (2). In the Extended Data figures this applies to: ED8f (3), ED8h (2), ED9b (2), ED9c (2), ED9d (2), ED9e (2), ED9f (5), ED9g (3), ED9h (2), ED10a (2), ED10b (2), ED10c (3), ED10d (2), ED10e (2), ED10f (2), ED10g (2), and ED10h (2). |
| Randomization | Random clones of the used yeast strains were selected for biochemical assays and mitochondrial isolations. The experiments from mitochondria were not randomized. All samples in one experiment were treated at the same time and with the same conditions. |
| Blinding | Blinding was not performed. The used yeast strains have to be validated before use. Thus, blinding is technically not feasible for the cell biological and biochemical assays performed. Also blinding is not required as all samples are treated in parallel. |

# Reporting for specific materials, systems and methods

We require information from authors about some types of materials, experimental systems and methods used in many studies. Here, indicate whether each material, system or method listed is relevant to your study. If you are not sure if a list item applies to your research, read the appropriate section before selecting a response.

## Materials & experimental systems

| n/a | Involved in the study |
|---|---|
| ☐ | ☒ Antibodies |
| ☐ | ☒ Eukaryotic cell lines |
| ☒ | ☐ Palaeontology and archaeology |
| ☒ | ☐ Animals and other organisms |
| ☒ | ☐ Human research participants |
| ☒ | ☐ Clinical data |
| ☒ | ☐ Dual use research of concern |

## Methods

| n/a | Involved in the study |
|---|---|
| ☒ | ☐ ChIP-seq |
| ☒ | ☐ Flow cytometry |
| ☒ | ☐ MRI-based neuroimaging |

# Antibodies

**Antibodies used**

Antibodies against proteins from baker's yeast Saccharomyces cerevisiae were generated in rabbits using peptides (Aco1 (GR947), Atp4 (GR1970), Cdc48 (GR5015), Cox2 (GR1948), Cox12 (GR1937), Cox14 (GR1544), Fis1 (GR310), Hsp60 (170), Ilv2 (GR1010), Mdh1 (GR1088), Mdj1 (121), Mdj2 (GR1842), mtHsp70 (GR119), Msp1 (GR1468), Mss51 (GR1952), Om14 (GR3041), Om45 (GR1311), Phb1 (298-11), Por1 (GR3621), Pth2 (GR797), Rsp5 (GR5064), Shy1 (GR1094), Tim23 (GR133), Tom7 (GR230), Ubp16 (GR5040), Ubx2 (GR1484), Yta10 (GR1550), Yta12 (GR1437), Pgk1 (GR753)) or recombinant proteins (Tom20 (GR3225), Tom22 (GR3227), Tom40 (168), Tom70 (GR657). The antisera were used in 1:250-1,000 dilution. Anti-DHFR (A9; Cat. sc-377091; dilution 1:1,000) and anti-ubiquitin (P4D1; Cat. sc-8017; dilution 1:1,000) antibodies were obtained from Santa Cruz Biotechnology. Anti-Pgk1 antibody (22C5D8, Cat. 459250; dilution 1:5,000) was obtained from Invitrogen. Secondary antibodies were obtained from Dianova (HRP-conjugated goat anti-rabbit, Cart. 111-035-003) or Li-Cor (goat anti-mouse IgG, IRDye 800CW, Cat. 926-32210; goat anti-rabbit IgG, IRDye 800CW, Cat. 926-32211; goat anit-mouse IgG, IRDye 680RD, Cat. 926-68070, goat anti-rabbit IgG, IRDye 680RD, Cat. 926-68071). Secondary antibodies were used at a concentration of 1:5,000 (HRP) or 1:10,000 (IRDye).

**Validation**

The specificity of the antibody raised against a protein from baker´s yeast (Saccharomyces cerevisiae) was controlled by comparing total cell extracts or mitochondrial lysates from wild-type yeast cells and the corresponding deletion strain or strains expressing a tagged version of the protein of interest via SDS-PAGE and Western blotting. Absence or size shift of the signal in cellular fractions of the mutant strain confirmed the specificity of the antibody signal. References for the used antibodies are:
Rabbit polyclonal anti-Aco1, Ref. 42
Rabbit polyclonal anti-Atp4, Ref. 69
Rabbit polyclonal anti-Cdc48, Ref. 17
Rabbit polyclonal anti-Cox2, Ref. 17
Rabbit polyclonal anti-Cox12, Ref. 42
Rabbit polyclonal anti-Cox14, Ref. 88
Rabbit polyclonal anti-Fis1, Ref. 21
Rabbit polyclonal anti-Hsp60, Ref. 42
Rabbit polyclonal anti-Ilv2, Ref. 100
Rabbit polyclonal anti-Mdh1, Ref. 101
Rabbit polyclonal anti-Mdj1, Ref. 17
Rabbit polyclonal anti-Mdj2, Ref. 42
Rabbit polyclonal anti-mtHsp70, Ref. 42
Rabbit polyclonal anti-Msp1, Ref. 17
Rabbit polyclonal anti-Mss51, Ref. 42
Rabbit polyclonal anti-Om14, Ref. 17
Rabbit polyclonal anti-Om45, Ref. 17
Rabbit polyclonal anti-Phb1, Ref. 102
Rabbit polyclonal anti-Por1, Ref. 21
Rabbit polyclonal anti-Pth2, Ref. 21
Rabbit polyclonal anti-Rsp5, this study
Rabbit polyclonal anti-Shy1, Ref. 42
Rabbit polyclonal anti-Tim23, Ref. 17
Rabbit polyclonal anti-Tom7, Ref. 80
Rabbit polyclonal anti-Tom20, Ref. 69
Rabbit polyclonal anti-Tom22, Ref. 69
Rabbit polyclonal anti-Tom40, Ref. 69
Rabbit polyclonal anti-Tom70, Ref. 69
Rabbit polyclonal anti-Ubp16, Ref. 21
Rabbit polyclonal anti-Ubx2, Ref. 17
Rabbit polyclonal anti-Yta10, Ref. 6
Rabbit polyclonal anti-Yta12, this study
Rabbit polyclonal anti-Pgk1, Ref. 17
mouse-monoclonal anti-DHFR, Santa Cruz, sc-377091
mouse-monoclonal anti-Pgk1, Invitrogen, 459250
mouse-monoclonal anti-ubiquitin, Santa Cruz, sc-8017

# Eukaryotic cell lines

Policy information about cell lines

**Cell line source(s)**

All yeast strains used in this study are described in Supplementary Table 4.
The yeast strains BY4741, pth2Δ, ubx2Δ, ubp16Δ, rsp5-1, mdm30Δ, mfb1Δ, and vms1Δ were obtained from EUROSCARF.

The yeast strains YPH499, Cox4-His, Tom20-His, Tom40-HA, ubx2Δ Tom40-HA, Ubx2-HA, pam17Δ, and pre9Δ pam17Δ have been described:

Sikorski, R. S. & Hieter, P. A system of shuttle vectors and yeast host strains designed for efficient manipulation of DNA in Saccharomyces cerevisiae. Genetics 122, 19-27 (1989).

Böttinger L. et al., A complex of Cox4 and mitochondrial Hsp70 plays an important role in the assembly of the cytochrome c oxidase. Mol. Biol. Cell 24, 2609-2619 (2013).

Mårtensson, C. U. et al. Mitochondrial protein translocation-associated degradation. Nature 569, 679-683 (2019).

Doan, K. N. et al. The mitochondrial import complex MIM functions as main translocase for α-helical outer membrane proteins. Cell Rep. 31, 107567 (2020).

The following strains were newly generated for this study: Phb1-ProtA, Tom22His Tom40-Strep, Pth2-His, pth2Δ Tom40-HA, pth2Δ pam17Δ, ubx2Δ pth2Δ,  ubp16Δ pth2Δ, pth2Δ vms1Δ, dsk2Δ, dsk2Δ pam17Δ, rsp5-1 ubp16Δ, rsp5-1 pam17, ubp16 pam17, mdm30Δ pam17Δ, mfb1Δ pam17Δ, hrd1Δ pam17Δ, doa10Δ pam17Δ, pth2Δ pam17Δ, pth2Δ pam17Δ + prs416, pth2Δ pam17Δ + prs416 Pth2, pth2Δ pam17Δ + prs416, pth2Δ pam17Δ + prs416 Pth2 D174A, pth2Δ pam17Δ + prs416, pth2Δ pam17Δ + prs416 Pth2 ΔTM, pth2Δ vms1Δ + prs416, pth2Δ vms1Δ + prs416 Pth2, pth2Δ vms1Δ + prs416 Pth2 D174A and pth2Δ vms1Δ + prs416 Pth2 ΔTM, tom70Δ, tom70Δ Tom40-HA.

Authentication

Yeast strains were grown on selective media for several generations. Deletion or tagging of genes was confirmed by western blot analysis of total cell extracts using the corresponding antibodies.

Mycoplasma contamination

Contamination with Mycoplasma is not an issue in yeast culturs and was therefore not tested.

Commonly misidentified lines
(See ICLAC register)

Commonly misidentified lines were not used in this study.

