## [Peer Review File · Nature]

Manuscript Title: Mitochondrial complexome reveals quality control pathways of protein import

Reviewer Comments & Author Rebuttals

Reviewer Reports on the Initial Version:

Referees' comments:

Referee #1 (Remarks to the Author):

This is an impressive paper describing a novel method for quantification of mitochondrial proteins and protein complexes. Using this technique the authors demonstrate that it is possible to capture protein complexes that had not been identified earlier. In the study yeast is used as a model organism, which enables the authors to leverage on previous extensive studies on this organism. I do not have any major comments on the paper, but it could benefit from more streamlining and making it more clear on the objective. E.g. 1) describe the method better in the introductory part and also the different experimental conditions that are compared; 2) make it more clear what the key findings are.

A few minor comments:

- 1) On page 6 it says that SDH subunit 3 co-assembles into SDH as well as TIM22. This could obviously be due to transport through TIM22 as also mentioned later, but this dynamic aspect is not made clear here.
- 2) I was wondering if the authors looked at the relative abundance of the various components of the ATPase complex? This could be interesting as it may hint towards assembly of this complex.

Referee #2 (Remarks to the Author):

Schulte and colleagues developed a novel approach to identify mitochondrial protein complexes by combing blue-native PAGE, a cryo-microtome to excise ~250 0.3mm gel slices, and mass spectroscopy to identify the protein constituents. Quite impressively, the authors identify ~5200 protein complexes located throughout all mitochondrial compartments as well as mitochondrial proteins that interact with cytosolic proteins linked to mitochondrial protein import, protein synthesis and protein quality control at the import channel.

In short, the findings are very impressive and will provide a useful resource to scientists in diverse fields. Surprisingly, on average each mitochondrial protein was found to localize within 6 or more independent protein complexes. Included in the mitochondrial protein complexes were complexes already well documented (MICOS), but numerous novel complexes were also revealed. The authors used their dataset, to address or explore a few unresolved, or controversial findings related to mitochondrial protein import and import quality control. For example, their data suggest that that Om14 (not Om45) directly links the TOM channel and a ribosome during mitochondrial protein import. They also identify the DNAJ protein that interacts with the complex comprised of prohibitin

and Yta10/12, which was known in mammals, but not yeast.

Lastly, the authors used the dataset to gain considerable insight into the processes that resolve mitochondrial precursor protein accumulation or import stalling, at the mitochondrial outer membrane or within the TOM channel. The mitochondrial outer membrane-localized peptidyl-tRNA hydrolase Pth2 was found to associate with the TOM complex and to be required to cleave mitochondrial precursor proteins that fail to be imported. Interestingly, precursor proteins are ubiquitylated by the ubiquitin ligase RSP5 if import is impaired. The authors also identify Dsk2 as being required for trafficking of ubiquitylated mito precursor proteins to the proteasome for degradation.

In conclusion, the development of a large and accessible dataset combined with the demonstration that the dataset can be used to address interesting biological questions will likely make the findings of broad interest.

Minor concern.

-The ubiquitin ligase RSP5 is suggested to be involved in ubiquitylating proteins that stall during mitochondrial protein import. RSP5 is documented to regulate numerous cellular processes including endocytic events. Most ubiquitylation by RSP5 requires a -PPxY- motif in the substrate proteins, which in turn leads to degradation via proteasomes. Does RSP5 ubiquitylate stalled mitochondrial import substrates by recognizing a similar motif or is it less specific in this context?

Referee #3 (Remarks to the Author):

Review of Nature 2022-05-07675, "High-resolution complexome of mitochondria reveals quality control pathways of protein import", by Pfanner and colleagues.

The authors use blue native gel electrophoresis coupled with quantitative mass spectrometry strategy to examine the composition of protein complexes in *S. cerevisiae* mitochondrial isolates. They derive a set of mitochondrial assemblies of yeast, termed 'MitCOM'. Of the 906 mitochondrial proteins, they discuss assemblies involved in respiration, metabolism, biogenesis etc. Quality control factors and their interactors predicted by MitCOM are further investigated. The data shows good correlation for well-established multiprotein structures, and the online tool allows users to select proteins of interest and examine co-fractionation on the gel samples. The subunits of expected large protein complexes showed similar profiles, highlighted by the authors such as MICOS, except the mitochondrial ribosome, which is no doubt a highly dynamic structure with many sub-structures. The follow-up validation experiments support some level of biological novelty and their findings will likely be of interest to members of the mitochondrial research community, but the overall interactome coverage is not substantially greater than what has already been published to date in yeast, including the references cited in the manuscript as well as the paper published by the same group in *Molecular Cell Proteomics* in 2016 using rat brain. The technique in itself is not conceptually different either, except the fact that it is applied to yeast rather than another model system, and not clear if any practical or technical improvements in the workflow. Based on this and the other concerns below, I am not convinced that the current data has substantially advanced the field,

beyond the detection of mitochondrial protein assemblies that are claimed to be of highest resolution and coverage, which would be better suited to report in a more targeted journal as opposed to claiming this study as a global resource.

Major comments:

1. How did the authors deconvolute and score the co-eluting peaks? How to they assign confidence to their predictions, or control for false discovery? While the authors validate several of the complexes in the manuscript, where did these complexes lie in the overall distribution of scores for all complexes? Complexes like the MICOS, which is actually 2 sub-complexes and an extra protein, came out as a single unit. Authors should comment that large structures rather than smaller substructures or subcomplexes are seemingly preferentially identified by this method.

2. While space is limited, I would like to see some further discussion of non-mitochondrial proteins. Almost half of the identified proteins are non-mitochondrial and would conceivably represent a large number of proteins which could, presumably, co-localize at some sort of contact sites. Despite being about half of the detected proteins, no significant comment is made as to how these proteins behaved overall.

3. In what way has this study exceeded previous complexome profiling/mapping efforts reported for mitochondria? Some of the complexes and interaction partners discussed in this manuscript have been well studied by this same group and others over the years. Discussion on the significance of the results, particularly on the dynamic organization of mitochondrial protein assemblies, should emphasize novel findings as opposed to those that are already known. For example, is Shy1 co-migrating with the respiratory super complex new? Peter Rehling's group in 2007 showed that Shy1 couples Cox1 translational regulation to cytochrome c oxidase assembly, and linked Shy1 an assembly factor for complex IV in *Saccharomyces cerevisiae*. In another example, with regards to Pth2 involvement in the removal of mitochondrial precursor proteins, the same group reported before in a Cell Reports paper that the import of Pth2 was reduced in tom70Δ mitochondria. Though some of the follow-up findings extend from the initial observations (e.g., Rsp5-UBP is a known connection but not with UBP16), these are consistent with a mechanistic follow up paper. Page 8, line5: The authors need to provide evidence for their claim that quality control pathways have components of low abundance.

Minor suggestions:

5. Why there is no loading control in several of the western blots? E.g., Extended Figure 6b, Extended Figure 8b, Extended Figure 8c?

6. The webtool is easy to use, though I did find it difficult to search for a specific protein without having to scroll down the page to find the desired protein alphabetically. On a similar note, I am disappointed by the presentation of the protein complex prediction. For example, one would like to examine possible co-fractionation of uncharacterized proteins, yet was unable to find similar profiles for unannotated proteins with known complexes without clicking randomly at known complex components until finding a candidate. This makes a webtool that is good at confirming a complex or interaction once suspected, but not at being used to assign novel function or make hypotheses.

7. A short methodological description of csBN-MS workflow is needed to help readers quickly grasp the technique. Since complexes in BN-PAGE are migrating through pores, is the separation linear/predictable over all ranges of complex compositions, number of proteins, shape, size, etc.?

Author Rebuttals to Initial Comments:

Referees' comments and point-by-point response

General

Based on the points raised by the Editor and the three referees, we have extensively revised our manuscript, including novel experimental data, further evaluation of the MitCOM dataset, and textual changes to better highlight the significance of the paper and the advance provided compared to previous work, as well as a more detailed description of the methods and approaches used. Moreover, we have improved the presentation and introduction of the interactive online platform for straightforward use by the readers (<https://www.complexomics.org/datasets/mitcom>).

We include the following new figures: Figs. 5e, 5f; Extended Data Figs. 1, 4a,b, 5, 7a-e, 8d, 9a, 9c; additional panels in Extended Data Figs. 9d, 9e, 10b, 10c; and additional columns in Supplementary Table 3.

Referee #1

This is an impressive paper describing a novel method for quantification of mitochondrial proteins and protein complexes. Using this technique the authors demonstrate that it is possible to capture protein complexes that had not been identified earlier. In the study yeast is used as a model organism, which enables the authors to leverage on previous extensive studies on this organism.

We are grateful to the referee for the constructive comments that helped to improve our manuscript.

I do not have any major comments on the paper, but it could benefit from more streamlining and making it more clear on the objective. E.g. 1) describe the method better in the introductory part and also the different experimental conditions that are compared; 2) make it more clear what the key findings are.

Following the suggestions of the referee, we systematically worked on streamlining the manuscript and presenting the objective, experimental conditions and findings more clearly. In addition, to better explain the methods, we include the new Extended Data Fig. 1 that provides an overview of the experimental workflow and the analysis of the MitCOM dataset. We present the key findings on the complexity of mitochondrial protein organization and the identification of novel interactors and quality control factors more clearly in the Main Text and Conclusions.

A few minor comments:

1) On page 6 it says that SDH subunit 3 co-assembles into SDH as well as TIM22. This could obviously be due to transport through TIM22 as also mentioned later, but this dynamic aspect is not made clear here.

We thank the reviewer for pointing this out. We now explain in the revised manuscript (and cite the relevant literature) that the precursor of Sdh3 is imported and inserted into the inner membrane via the presequence translocase TIM23 and the OXA translocase, not via the TIM22 complex. Mature Sdh3 subsequently assembles into the SDH complex

and the TIM22 complex. Due to space restrictions, we include this detailed explanation in the legend of Extended Data Fig. 8b, c, where the data on Sdh3 are shown.

2) I was wondering if the authors looked at the relative abundance of the various components of the ATPase complex? This could be interesting as it may hint towards assembly of this complex.

As suggested by the referee, we include the new Extended Data Fig. 8d that presents a detailed analysis of the F₁F₀-ATP synthase, revealing the major dimeric form, the monomeric form, a primed monomer with dimerization factors, the F₁-module, and a close co-migration of Atp1 and Atp2 in the low molecular mass range. We also point out in the revised manuscript that the predominant presence of the physiological, fully assembled dimer of the ATP synthase in the blue native separation underscores our optimized conditions for a mild and efficient lysis of the mitochondrial membranes (in the usual conditions for membrane lysis and blue native separation, dimer and monomer of the ATP synthase are typically present in similar abundance).

Referee #2

Schulte and colleagues developed a novel approach to identify mitochondrial protein complexes by combing blue-native PAGE, a cryo-microtome to excise ~250 0.3mm gel slices, and mass spectroscopy to identify the protein constituents. Quite impressively, the authors identify ~5200 protein complexes located throughout all mitochondrial compartments as well as mitochondrial proteins that interact with cytosolic proteins linked to mitochondrial protein import, protein synthesis and protein quality control at the import channel.

In short, the findings are very impressive and will provide a useful resource to scientists in diverse fields. Surprisingly, on average each mitochondrial protein was found to localize within 6 or more independent protein complexes. Included in the mitochondrial protein complexes were complexes already well documented (MICOS), but numerous novel complexes were also revealed. The authors used their dataset, to address or explore a few unresolved, or controversial findings related to mitochondrial protein import and import quality control. For example, their data suggest that that Om14 (not Om45) directly links the TOM channel and a ribosome during mitochondrial protein import. They also identify the DNAJ protein that interacts with the complex comprised of prohibitin and Yta10/12, which was known in mammals, but not yeast.

Lastly, the authors used the dataset to gain considerable insight into the processes that resolve mitochondrial precursor protein accumulation or import stalling, at the mitochondrial outer membrane or within the TOM channel. The mitochondrial outer membrane-localized peptidyl-tRNA hydrolase Pth2 was found to associate with the TOM complex and to be required to cleave mitochondrial precursor proteins that fail to be imported. Interestingly, precursor proteins are ubiquitylated by the ubiquitin ligase RSP5 if import is impaired. The authors also identify Dsk2 as being required for trafficking of ubiquitylated mito precursor proteins to the proteasome for degradation.

In conclusion, the development of a large and accessible dataset combined with the demonstration that the dataset can be used to address interesting biological questions will likely make the findings of broad interest.

We are grateful to the referee for the constructive comments that helped to improve our manuscript.

Minor concern.

-The ubiquitin ligase RSP5 is suggested to be involved in ubiquitylating proteins that stall during mitochondrial protein import. RSP5 is documented to regulate numerous cellular processes including endocytic events. Most ubiquitylators by RSP5 requires a -PPxY- motif in the substrate proteins, which in turn leads to degradation via proteasomes. Does RSP5 ubiquitylate stalled mitochondrial import substrates by recognizing a similar motif or is it less specific in this context?

Following the suggestion of the referee, we have addressed this question and include the new Figs. 5e, 5f and Extended Data Fig. 9c in the revised manuscript. We now point out that Rsp5 preferentially binds to substrates and adaptor proteins via interaction with a PPxY/LPxY motif and cite the relevant literature. The receptor Tom70 indeed contains a PPxY motif and may thus be involved in recruiting Rsp5 to mitochondria. We now show that the interaction of Rsp5 with the TOM complex was impaired in the absence of Tom70 and precursor ubiquitylation was diminished (Fig. 5e, f). In addition, we found an LPxY motif in the DHFR domain of the precursor protein *b*₂-DHFR and show that the ubiquitylation of a modified precursor (GGxY) was reduced (Extended Data Fig. 9c). These data suggest that Tom70 is involved in recruiting Rsp5 to the mitochondrial import site. Rsp5 may use Tom70 as an adapter for interacting with precursor proteins and, in case a precursor substrate contains a PPxY/LPxY motif, also engage the motif on the precursor.

Referee #3

*The authors use blue native gel electrophoresis coupled with quantitative mass spectrometry strategy to examine the composition of protein complexes in *S. cerevisiae* mitochondrial isolates. They derive a set of mitochondrial assemblies of yeast, termed 'MitCOM'. Of the 906 mitochondrial proteins, they discuss assemblies involved in respiration, metabolism, biogenesis etc. Quality control factors and their interactors predicted by MitCOM are further investigated. The data shows good correlation for well-established multiprotein structures, and the online tool allows users to select proteins of interest and examine co-fractionation on the gel samples. The subunits of expected large protein complexes showed similar profiles, highlighted by the authors such as MICOS, except the mitochondrial ribosome, which is no doubt a highly dynamic structure with many sub-structures. The follow-up validation experiments support some level of biological novelty and their findings will likely be of interest to members of the mitochondrial research community, but the overall interactome coverage is not substantially greater than what has already been published to date in yeast, including the references cited in the manuscript as well as the paper published by the same group in *Molecular Cell Proteomics* in 2016 using rat brain. The technique in itself is not conceptually different either, except the fact that it is applied to yeast rather than another model system, and not clear if any practical or technical improvements in the workflow. Based on this and the other concerns below, I am not convinced that the current data has substantially advanced the field, beyond the detection of mitochondrial protein assemblies that are claimed to be of highest resolution and coverage, which would be better suited to report in a more targeted journal as opposed to claiming this study as a global resource.*

We are grateful to the referee for the constructive comments that helped to improve our manuscript.

As outlined below in the response to the individual points of this referee, we compared the MitCOM dataset with the study of Müller et al. (2016 *Molecular & Cellular*

Proteomics, ref. 10) that reported the so far highest resolution of a mitochondrial complexome. (i) Müller et al. resolved ~2,500-2,600 protein peaks and thus only about 50% of the overall interactome coverage of MitCOM (5,224 protein peaks). (ii) Importantly, since Müller *et al.* lacked the advanced data processing of MitCOM, the possibilities for discovering novel interactors were strongly limited in contrast to MitCOM. (iii) Other mitochondrial complexome studies used considerably smaller numbers of slices per blue native gel separation and thus have a substantially lower interactome coverage (detailed in major point 3 below).

Following the suggestion of the referee, we included the new Extended Data Fig. 1 that outlines the experimental workflow for the determination, analysis and evaluation of MitCOM and highlights novel or significantly improved tools in red.

Based on the suggestions of referees 1 and 3 we present the key findings on the complexity of mitochondrial protein organization and the identification of novel interactors and quality control factors more clearly in the revised manuscript.

Major comments:

1. How did the authors deconvolute and score the co-eluting peaks? How to they assign confidence to their predictions, or control for false discovery? While the authors validate several of the complexes in the manuscript, where did these complexes lie in the overall distribution of scores for all complexes?

As suggested by the referee, we included the new Extended Data Fig. 1, Fig. 4 and Fig. 5 and substantially improved the description of the methods and evaluation tools used. Extended Data Fig. 1 outlines the experimental workflow for the determination, analysis and evaluation of MitCOM and highlights novel or significantly improved tools in red. Co-eluting peaks were inspected using the ProfilViewer tool with the local correlation function (Extended Data Fig. 3c), providing pairwise correlation coefficients. The reliability of the procedure was emphasized by MitCOM-wide correlation analyses starting out from Gaussian-fitted profile peaks. A distance measure combining information on peak intensity, mass range and correlation coefficients was used for automated discrimination of protein complex components by t-distributed stochastic neighborhood distribution (t-SNE) (shown in Extended Data Figs. 4 and 5 and detailed in Methods).

Extended Data Fig. 4a shows a histogram of similarity scores obtained for the segments of the abundance-mass profiles of MitCOM proteins (total of 1.2×10^9 values). The scores integrate (i) correlation of the profile segments of any MitCOM protein with those from all MitCOM proteins (local Pearson correlation), (ii) ratio of their molecular abundance and (iii) distance of their maxima (now detailed in Methods). A value of 1.0 indicates a perfect agreement of all three parameters. As requested by the referee, in Extended Data Fig 4b we present the pairs in the groups/complexes of protein profile peaks that are analyzed in this study, revealing that they have similarity score values between 1.0 - 0.88.

Finally, we clearly point out in the revised manuscript (Main text, Conclusions and Extended Data Fig. 1) that co-migrating protein peaks (potential novel interactors) have to be independently verified biochemically by affinity purification and functional analysis

and show examples in the Figs. 3, 4, 5 and 6 (complemented by Extended Data Figs. 9 and 10).

Major comment 1 continued:

Complexes like the MICOS, which is actually 2 sub-complexes and an extra protein, came out as a single unit. Authors should comment that large structures rather than smaller substructures or subcomplexes are seemingly preferentially identified by this method.

We thank the referee for the comment. In the revised manuscript we explain in detail that we used optimized conditions for a mild and efficient lysis of the mitochondrial membrane preparation by applying 8 mg purified digitonin to 1 mg mitochondrial protein (detailed in Methods and outlined in Main text and Extended Data Fig. 1). We found that the protein-digitonin ratio of 1:8 allows for an efficient extraction of yeast mitochondrial membrane protein complexes, yet is substantially milder than the 1:10 protein-digitonin ratio that is typically used for yeast mitochondria or the application of other detergents. This optimized mild solubilization condition is illustrated by the predominant presence of the physiological, fully assembled dimer of the F₁F₀-ATP synthase in the blue native separation (new Extended Data Fig. 8d), whereas the typical 1:10 ratio conditions lead to an about equal distribution between ATP synthase dimer and monomer on blue native gels (relevant literature cited in the manuscript). It is thus likely that the predominant migration of the MICOS complex as very large, single unit is due to our optimized conditions for a mild and efficient lysis of the mitochondrial membranes that is likely closer to the physiological *in organello* situation than the harsher conditions previously used. We envisage that our observation on the size and migration behavior of MICOS will stimulate interesting future studies on the molecular organization of MICOS and its role in stabilizing the inner membrane crista junctions of mitochondria.

These mild conditions for membrane lysis are likely responsible for our observation that larger structures of protein complexes are retained during lysis and blue native separation, allowing for the detection of supercomplexes that are otherwise dissociated by harsher solubilization conditions. However, it is important to point out (and explained in the manuscript) that the resolution of MitCOM is homogenous over the large molecular mass range from ~80 kDa to ~3,800 kDa (Fig. 1c, d, Extended Data Fig. 2b) and that MitCOM faithfully detects smaller complexes (e.g., Figs. 2, 3a, 3e, 4a, 5a, Extended Data Figs. 6c, 8a, 8b, 8c and 8d).

2. While space is limited, I would like to see some further discussion of non-mitochondrial proteins. Almost half of the identified proteins are non-mitochondrial and would conceivably represent a large number of proteins which could, presumably, co-localize at some sort of contact sites. Despite being about half of the detected proteins, no significant comment is made as to how these proteins behaved overall.

Following the suggestion of the referee, we include the new Extended Data Fig. 7a-e that summarizes the complexity, localization and functional classification of the non-mitochondrial proteins. These proteins showed an average complexity of 4.5 peaks per protein and predominantly originated from cellular compartments in the vicinity of mitochondria: cytosol, endoplasmic reticulum and nucleus. 'Protein biogenesis, degradation and quality control' represented the largest functional class of non-mitochondrial proteins, consistent with the identification of quality control factors

associated with mitochondria. In addition, we extended the Supplementary Table 3 and included the information on subcellular localization, predicted transmembrane domains and functional classification of the non-mitochondrial proteins, in addition to the detailed information related to MS identification and quantification.

3. In what way has this study exceeded previous complexome profiling/mapping efforts reported for mitochondria?

We compared the MitCOM dataset with the study by Müller *et al.* (2016 *Molecular & Cellular Proteomics*, ref. 10) that reported the so far highest resolution of a mitochondrial complexome (source: rat brain). The number of 230 slices generated from the blue native gel in Müller *et al.* represents 94% of the top number of 245 slices of MitCOM. Other published mitochondrial complexome studies use <100 slices per blue native gel separation (96, 64 or less slices) and thus <40% of the slices used by MitCOM (summarized in Wittig & Malacarne, 2021/ref. 8, and the CEDAR repository: <https://www3.cmbi.umcn.nl/cedar>).

Müller *et al.* (2016), however, lacks the high precision provided by the advanced MitCOM data processing that is a prerequisite for a reliable automated peak fitting and correlation analysis (detailed in Extended Data Fig. 1 and Methods). As a rough estimate, the protein profiles from Müller *et al.* showed in average only 3-4 reliably fittable peaks per protein, compared to the average of 6.4 peaks per MitCOM protein that can be directly determined by the unbiased automated component analysis based on peak-detection and fitting of multi-Gaussian functions to the abundance-mass profiles. In addition, the proteome coverage of Müller *et al.* is substantially lower (~65% of the expected mitochondrial proteome size) than that of MitCOM (>90% coverage).

Taken together, we estimate that the study by Müller *et al.* (2016) separates ~2,500-2,600 protein peaks and thus covers only ~50% of the mitochondrial complexity of MitCOM with 5,224 protein peaks. Importantly, since Müller *et al.* lack the advanced possibilities for automated peak fitting, correlation analysis and searching for partner proteins, the possibilities for discovering novel interactors are quite limited in contrast to the immense and unprecedented possibilities of MitCOM in discovering novel interactors of mitochondrial protein complexes. In addition, Müller *et al.* does not contain the functional characterization of novel interactors as performed with the quality control factors in the current study. Other mitochondrial complexome studies use <40% of the number of slices and thus their separation and resolution are substantially lower and do not allow for the detailed comparison of the shapes of co-migrating proteins and the efficient identification of potential protein interactors as in MitCOM.

Major comment 3 continued:

Some of the complexes and interaction partners discussed in this manuscript have been well studied by this same group and others over the years. Discussion on the significance of the results, particularly on the dynamic organization of mitochondrial protein assemblies, should emphasize novel findings as opposed to those that are already known.

We thank referee 3 and referee 1 for pointing this out. We now discuss the novel findings and their significance with regard to the dynamic organization of mitochondrial protein assemblies, the complexity of mitochondrial protein organization, and the identification

of novel interactors and quality control factors more clearly in the Main Text and Conclusions, whereas verification studies and details on known complexes are presented in Extended Data and the corresponding figure legends.

Major comment 3 continued:

For example, is Shy1 co-migrating with the respiratory super complex new? Peter Rehling's group in 2007 showed that Shy1 couples Cox1 translational regulation to cytochrome c oxidase assembly, and linked Shy1 an assembly factor for complex IV in Saccharomyces cerevisiae.

We have now clarified in the revised manuscript that for Shy1 the quantitative nature of MitCOM is the important novel point. Qualitative blue native studies suggested that only a small fraction of Shy1 is associated with the full III₂IV₂ supercomplex (mentioned and relevant literature cited in the first version of the manuscript). Due to the presumed small amount, however, possible functional implications of this association were not considered further. MitCOM provides a direct quantitative analysis and demonstrates that a main peak of Shy1 is present at the fully assembled supercomplex, different from other assembly factors such as Mss51 and Coa1. We envisage that this quantitative finding on Shy1 may stimulate future research on the role of the assembly factor Shy1 at the full supercomplex.

Major comment 3 continued:

In another example, with regards to Pth2 involvement in the removal of mitochondrial precursor proteins, the same group reported before in a Cell Reports paper that the import of Pth2 was reduced in tom70Δ mitochondria. Though some of the follow-up findings extend from the initial observations (e.g., Rsp5-UBP is a known connection but not with UBP16), these are consistent with a mechanistic follow up paper.

We now describe more clearly in the revised manuscript that the **precursor** of Pth2 is imported from the cytosol via the receptor Tom70 and inserted into the outer membrane by the mitochondrial import complex MIM (reported in our Cell Reports paper Doan *et al.*, 2020, ref. 41). Since Tom70 is a major mitochondrial protein import receptor, a large number of mitochondrial precursor proteins from several mitochondrial subcompartments are imported via Tom70; in nearly all cases, the mature, fully imported proteins do not interact with the import receptor anymore. It has been unknown whether mature Pth2 has interaction partners in the mitochondrial outer membrane and whether Pth2 plays a role in mitochondrial function or quality control (and this is now pointed out more clearly in the revised manuscript). Based on the MitCOM dataset and the biochemical and functional analysis shown in our study, we discovered that Pth2 interacts with the TOM complex and constitutes a novel pathway for the removal of accumulated precursor proteins from the mitochondrial entry gate. We also point out that neither an E3 ubiquitin ligase nor a deubiquitylase operating at the yeast TOM complex had been identified. Using the MitCOM dataset we were able to identify Rsp5 and Ubp16 as responsible enzymes at TOM and combined with the functional analysis we can present the new Pth2 quality control pathway together with relevant interacting factors.

Page 8, line5: The authors need to provide evidence for their claim that quality control pathways have components of low abundance.

We include the new Extended Data Fig. 9a that presents the copy numbers per yeast cell for TOM proteins and quality control factors operating at TOM (based on the absolute quantification provided in the yeast proteome study by Morgenstern *et al.*, 2017, ref. 7). This figure clearly shows that the quality control factors are of considerably lower abundance than the TOM subunits under three different growth conditions (glucose, galactose or glycerol as carbon source).

Minor suggestions:

5. Why there is no loading control in several of the western blots? E.g., Extended Figure 6b, Extended Figure 8b, Extended Figure 8c?

We have included the loading controls as requested by the referee in the Extended Data Figs. 9d, 9e, 10b, and 10c (the figures were renumbered in the revised manuscript).

6. The webtool is easy to use, though I did find it difficult to search for a specific protein without having to scroll down the page to find the desired protein alphabetically. On a similar note, I am disappointed by the presentation of the protein complex prediction. For example, one would like to examine possible co-fractionation of uncharacterized proteins, yet was unable to find similar profiles for unannotated proteins with known complexes without clicking randomly at known complex components until finding a candidate. This makes a webtool that is good at confirming a complex or interaction once suspected, but not at being used to assign novel function or make hypotheses.

Following the suggestions of the referee, we have thoroughly revised the webtool and improved its presentation and usability. We included more detailed descriptions on using the webtool in Methods and the legend of Extended Data Fig. 3 and integrated a detailed Help function on the WEB-Site (<https://www.complexomics.org/datasets/mitcom>) to explain the use of the search function also for protein complex prediction, providing the top 20 co-migrating protein peaks for any MitCOM protein peak selected.

7. A short methodological description of csBN-MS workflow is needed to help readers quickly grasp the technique. Since complexes in BN-PAGE are migrating through pores, is the separation linear/predictable over all ranges of complex compositions, number of proteins, shape, size, etc.?

We have extended the description of the approach in Methods and provided a detailed workflow in the new Extended Data Fig. 1. The separation power of our BN-PAGE was homogenous on the entire range from ~80 kDa to ~3.8 MDa (Fig. 1c, d, Extended Data Fig. 2b). Slight deviations from linearity are obvious from the mass-size calibration approximated by a sigmoidal function fitted to complexes with established mass (Extended Data Fig. 2c). In addition, we now cite the comprehensive study by Wittig *et al.* (2010, ref. 75) and point out that this study provides a detailed discussion of the principles of mass estimation of protein complexes by blue native electrophoresis.

Author Rebuttals to First Revision:

Referees' comments:

Referee #1 (Remarks to the Author):

I think the authors have addressed all reviewer comments well and the revised manuscript is much improved

Referee #2 (Remarks to the Author):

The authors have addressed my concerns. I support publication of the manuscript.

Referee #3 (Remarks to the Author):

The authors have addressed my specific main concerns.